# Identifying critical growth stage and resilient genotypes in cowpea under drought stress contributes to enhancing crop tolerance for improvement and adaptation in Cameroon

**Toscani Ngompe Deffo, Eric Bertrand Kouam** **\*, Marie Solange Mandou, Raba Allah-To Bara, Asafor Henry Chotangui, Adamou Souleymanou, Honore Beyegue Djonko, Christopher Mubeteneh Tankou**

Genetics, Biotechnology, Agriculture and Plant Production Research Unit, Department of Crop Science, Faculty of Agronomy and Agricultural Science, University of Dschang, Dschang, Cameroon

\* ericbkouam@yahoo.com, eric.kouam@univ-dschang.org

## Abstract

Drought stress following climate change is likely a scenario that will have to face crop growers in tropical regions. In mitigating this constraint, the best option should be the selection and use of resilient varieties that can withstand drought threats. Therefore, a pot experiment was conducted under greenhouse conditions at the Research and Teaching Farm of the Faculty of Agronomy and Agricultural Sciences of the University of Dschang. The objectives are to identify sensitive growth stage, to identify drought-tolerant genotypes with the help of yield-based selection indices and to identify suitable selection indices that are associated with yield under non-stress and stress circumstances. Eighty-eight cowpea genotypes from the sahelian and western regions of Cameroon were subjected to drought stress at vegetative (VDS) and flowering (FDS) stages by withholding water for 28 days, using a split plot design with two factors and three replications. Seed yields under stress (Ys) and non-stress (Yp) conditions were recorded. Fifteen drought indices were calculated for the two drought stress levels against the yield from non-stress plants. Drought Intensity Index (DII) under VDS and FDS were 0.71 and 0.84 respectively, indicating severe drought stress for both stages. However, flowering stage was significantly more sensitive to drought stress compared to vegetative stage. Based on PCA and correlation analysis, Stress Tolerance Index (STI), Relative Efficiency Index (REI), Geometric Mean Productivity (GMP), Mean Productivity (MP), Yield Index (YI) and Harmonic Mean (HM) correlated strongly with yield under stress and non-stress conditions and are therefore suitable to discriminate high-yielding and tolerant genotypes under both stress and non-stress conditions. Either under VDS and FDS, CP-016 exhibited an outstanding performance under drought stress and was revealed as the most drought tolerant genotype as shown by ranking, PCA and cluster analysis. Taking into account all indices, the top five genotypes namely CP-016, CP-021, MTA-22, CP-056 and CP-060 were identified as the most drought-tolerant genotypes under VDS. For stress activated at flowering stage (FDS), CP-016, CP-056, CP-021, CP-028 and MTA-22 were the top five most drought-tolerant genotypes. Several genotypes with insignificant Ys

**Data Availability Statement:** All relevant data from this research are available within the paper.

**Funding:** The author(s) received no specific funding for this work.

**Competing interests:** The authors have declared that no competing interests exist

and irrelevant rank among which CP-037, NDT-001, CP-036, CP-034, NDT-002, CP-031, NDT-011 were identified as highly drought sensitive with low yield stability. This study identified the most sensitive stage and drought tolerant genotypes that are proposed for genetic improvement of cowpea.

## Introduction

Cowpea, of scientific name *Vigna unguiculata* is an important food legume in developing countries, mostly in sub-Saharan Africa, Asia, Central and South America [1, 2]. Cowpea has been referred to as "poor man's meat" because of its high protein content, 20 to 25% [3, 4], as "hungry season crop" because, when grown together with cereals, it is the first crop to be harvested [5]. Owing to its capacity of fixing atmospheric nitrogen which can be up to 337 kgN. ha$^{-1}$ [6], cowpea tolerates low soil fertility and its inclusion in crop rotations makes it possible to meet the nitrogen fertilizer needs of subsequent crops [7]. This characteristic helps increasing the yields of crops and subsequently contributes to the sustainability of cropping systems [8–10]. Cowpea young leaves, pods and peas contain vitamins and minerals useful for human consumption and animal feeding [3, 11]. Sub-Saharan Africa accounts for more than 64% of the world's cowpea production, estimated to about 12.5 million tons [12]. In Cameroon the national production is estimated to about 110,000 tons from a planted area of 105,000 hectares [13]. At the regional level in Cameroon, cowpea is mostly produced in the Sahelian region, ordinarily for trade and feeding, and ranks second after groundnuts in the category of leguminous crops [14]. Grain yield in this region ranges from 300 to 500 kg/ha in farmers' fields and 1,200 to 2,000 kg/ha in research stations which use improved varieties [14]. Cowpea is also significantly produced in the Western region of the country, mostly for local consumption and cultural considerations [15].

Drought also known as water deficit is one of the abiotic factors limiting significantly the crop production [16]. Numerous reports qualify drought as one of the devastating environmental stress that decrease crop productivity [17, 18]. This moisture deficit stress condition seems to increase in arid and semi-arid regions and is likely to be highly considered for future climate change scenarios [19, 20]. In Cameroon, climate change is perceptible with gradual elevation of temperature. Key sectors to be exposed include plant and animal production [21]. Hussain et al. [22] and Moukoumbi et al. [23] underline the use of genotypic variability in crop under drought stress for selecting resilient genotype in order to fight climate change. Cowpea itself according to the literature, can withstand dry environmental conditions at some extent when compared to other crops [24]. This demonstrates its tolerance to water deficit conditions that could have been very disastrous to many other annual crops. In cowpea, report of Turk et a. [25] indicates that after a drought spell, resuming watering will lead to a yield that can be similar or different to that of normally irrigated plant, this depending on the growth stage where plants were exposed to drought stress. Cowpea is heavily damaged by repeated water deficits [26]. Some researchers like Benjamin et al. [27] and Demirevska et al. [28] pointed that crop failure often occurs because of the unavailability of water at some critical growth stages. In common bean, a related leguminous crop, drought stress during flowering stage results in 60 to 99% yield loss [29]. Early maturing cowpea genotypes are more susceptible to water deficit during flowering and fruiting phases [30]. Drought reduces cowpea potential from 1000 kg of grains/ha in normal condition, to around 360 kg/ha when stress sets-in during pre-flowering [31]. This drastic reduction is likely to be accounted to the unavailability

of high yielding varieties resistant to moisture stress [32, 33]. Yield and responses to drought stress differ between varieties in many crops. Pejic et al. [26] revealed significant difference among cowpea varieties when exposed to water deficit conditions.

In order to ensure food security for the increasing world population, the approach involving a better use of available water for the development of crop is to be highly considered. With climate change making the earth warmer, using varieties which needs less amount of water and more tolerance to drought will be the best option for crop production [34]. Yield loss is the main concern of plant breeders as farmers usually experience low yield performance under stress conditions. Therefore, drought indices which provide a measure of drought based on loss of yield under drought-conditions in comparison to normal conditions have been largely used for screening drought tolerant genotypes in many crops including cowpea [26], common bean [35–37], maize [38, 39], wheat [40] and sorghum [41]. According to Fernandez [42], the best indices are those having high correlation with grain yield in both stress and non-stress conditions and the best selection in drought condition should involve genotypes having desirable yield in stress and non-tress condition. Although, there are several reports on the association of the indices with drought tolerance in many crops, very few studies were carried out on cowpea and using limited number of genotypes. The evaluation of a large quantity of genotypes from various sources is the novelty of this study. This is particularly important for exploiting the available genetic variation and identifying the most tolerant genotypes that will be valued by cowpea breeders in developing superior cultivars. The objectives of this study were to assess eighty-eight cowpea genotypes under non stress and stress conditions activated at the vegetative and flowering stages in order to: (i) identify the sensitive growth stage in cowpea to drought stress by the help of selection indices, (ii) identify high-yielding drought tolerant genotypes using tolerance indices, (iii) identify suitable stress indices which are significantly correlated with seed yield under both non-stress and stress conditions for their use as quick identifiers in cowpea breeding for drought stress tolerance.

## Materials and methods

### Study site and soil characteristics

The experiment was carried out in the genetic experimental greenhouse at the Research and Teaching Farm of the Faculty of Agronomy and Agricultural Sciences of the University of Dschang, located in the Western Region of Cameroon at 5˚20' North latitude and 10˚05' East longitude, and 1407 m asl. Others characteristics of the study site include an average annual temperature of 21.50˚C and the relative humidity of 76.8%. The texture of the soil used in this experiment was sandy clay loam with 40%, 32% and 28% of sand, silt and clay, respectively. The electrical conductivity of soil was 60 $\mu$S cm$^{-1}$. The experimental soil contained 1.36 gkg$^{-1}$ organic matter and 0.02 gkg$^{-1}$ total nitrogen contents. Soil field capacity (FC) was 52.7% and permanent wilting point (PWP) was 26.3%. Plant available water (PAW) was estimated as follows: PAW = FC−PWP [43]

### Plant material and different water regimes

Eighty-eight cowpea accessions used for the study were collected from the Sahelian and the Western highland region of Cameroon. Accessions were from the Institute of Agricultural Research for Development (9 accessions) and 79 landraces originated from market places and farmers' fields. Three treatments namely, non-drought stress (NDS), drought stress at vegetative stage (VDS), and drought stress at flowering stages (FDS) were applied. Drought stress consisted at withholding water for 28 days at each of the corresponding stage.

## Experimental design and procedures

The experimental design was a split plot involving two factors with three replications. The main factor was the water regime with 3 levels: no stress = regular irrigation (NDS), stress at the vegetative stage (VDS, started at around three weeks from planting, when the first trifoliate leaf appeared open and flat on 50% of plants) and stress initiated and flowering stage (FDS, when the first open flower appeared on 50% of plants). The subplot consisted of genotypes, 88 in total and constituted the second factor. Plastic pots of 5 litres capacity (25 cm x 30 cm dimension) and perforated at the bottom were filled with top soil collected up to 30 cm depth around the experimental site. Five seeds of each cowpea genotype were planted in each pot, and two weeks after sowing, the seedlings were thinned to three seeds per pot and to two seeds per pot one weeks later. All pots were well watered with two-day interval frequency and up to the initiation of drought stress. One day before starting treatments, soil moisture in each pot was maintained to field capacity for the purpose of keeping uniform soil moisture at each pot. This generated 264 interactions repeated 3 times, giving 792 experimental units. Within an experimental unit, plants were spaced 0.3 m within rows and 0.5 m between rows. Replications were separated from each other by 0.5 m. For stressed plants, the protocol of Fernandez [42] was followed with the irrigation completely cut off for 28 days before resuming to normal irrigation. Under non-drought stress condition, plants were regularly watered up to field capacity until physiological maturity.

## Drought tolerance indices and data analysis

At harvest, seed were collected on each plant. These seeds were then weighted using pioneer sensitive digital balance and seed yield estimated. In addition to the direct measurement, some derived variables that are drought tolerance indices were calculated from primary data for the two drought stress stages from grain yield under the drought-stressed (VDS and (FDS), and non-stressed (NDS) conditions according to various formula that are indicated in Table 1. Seed yield data were recorded for each genotype at both stress and non-stress environment and were subjected to the estimation of the different drought selection indices following appropriate formula and using Microsoft Excel software. Mean separation using Student's t test was performed to compare VDS and FDS for the various drought indices. Pearson correlation analysis was performed using XLSTAT software (version 2016) and R (version 4.3.3) computer programs to determine the association between drought tolerance indices and grain yield in both stressed and non-stressed conditions. To determine the most desirable drought-tolerant genotype, rank mean value was given based on each selection indices rank value. This rank mean was calculated by averaging for all selection indices value for the considered genotype. The standard deviation (the deviation of each rank from the mean of rank) of rank mean was also calculated. The rank sum was calculated according to Yahaya et al. [41] using this relationship formula: $Rank\ Sum\ (RS) = Rank\ Mean\ (\bar{R}) + Standard\ Deviation\ of\ Rank (SDR)$. Genotypes with low rank mean and low standard deviation of rank were considered as best drought-tolerant. Besides ranking methods, Principal component and hierarchical clustering analysis from the XLSTAT software (Version 2016) were used for the identification of the suitable tolerant genotypes following Fernandez [42] recommendations. Cluster analysis was carried out using Ward methods with squared euclidean distances [44]

## Results

### Drought tolerance indices and sensitive growth stage

Compared to controls plants (NDS), plants exposed to VDS and FDS each gave a seed yield significantly lower with differential responses among genotypes. Between VDS and FDS,

**Table 1. Code, name and formula applied to estimate the fourteen drought tolerance indices from grain yield under the drought-stressed (Ys) and non-stressed conditions (Yp).**

| No | Index code | Index name | Formula | Remark | Reference |
|----|-----------|-----------|---------|--------|-----------|
| 1 | SSI | Stress Susceptibility Index | $SSI = \frac{1-\left(\frac{Ysi}{Ypi}\right)}{1-\left(\frac{Ys}{Yp}\right)}$ | The greater the index, the greater susceptibility of the genotype to stress. | [45] |
| 2 | RDI | Relative Drought Index | $RDI = \frac{\frac{Ysi}{Ypi}}{\frac{Ys}{Yp}}$ | the genotype is relatively drought tolerant if RDI > 1 and drought susceptible if RDI < 1 | [46] |
| 3 | SSPI | Stress Susceptibility Percentage Index | $SSPI = \frac{Ypi-Ysi}{2(Yp)}$ | The smaller SSPI, the more relative tolerance | [45] |
| 4 | SDI | Sensitivity Drought Index | $SDI = \frac{Ypi-Ysi}{Ypi}$ | The genotypes with small value of this index will be more desirable | [42] |
| 5 | TOL | Tolerance index | $TOL = Ypi - Ysi$ | The genotypes with small value of this index will be more desirable | [42] |
| 6 | STI | Stress Tolerance Index | $STI = \frac{Ypi \cdot Ysi}{(Yp)^2}$ | The genotypes with high STI values are more tolerant to drought stress | [42] |
| 7 | REI | Relative Efficiency Index | $REI = \frac{Ysi}{Ys} * \frac{Ypi}{Yp}$ | The genotypes with high value of this index will be more desirable | [47] |
| 8 | GMP | Geometric Mean Productivity | $GMP = \sqrt{Ypi * Ysi}$ | The higher GMP value, the greater the degree of relative tolerance | [42] |
| 9 | MP | Mean Productivity | $MP = \frac{Ypi+Ysi}{2}$ | The genotypes with high value of this index will be more desirable | [42] |
| 10 | YSI | Yield Stability Index | $YSI = \frac{Ysi}{Ypi}$ | The genotypes with high values can be regarded as stable genotypes under stress and non-stress conditions | [48] |
| 11 | YI | Yield Index | $YI = \frac{Ysi}{Ys}$ | The genotypes with high value of this index will be suitable for drought stress condition | [49] |
| 12 | PYR | Percentage of Yield Reduction | $PYR = \left(\frac{Ypi-Ysi}{Ypi}\right)*100$ | The genotypes with small value of this index will be more desirable | [50] |
| 13 | DTI | Drought Tolerance Index | $DTI = \frac{Ysi\left(\frac{Ysi}{Ypi}\right)}{Ys}$ | genotypes with the highest DTI will be desirable to moisture stress | [51] |
| 14 | HM | Harmonic Mean | $HM = \frac{2(Yp*Ys)}{Yp+Ys}$ | The genotypes with high value of this index will be more desirable | [52] |
| 15 | DII | Drought Intensity Index | $DII = 1 - \frac{Ys}{Yp}$ | The genotypes with low value of DII will be more desirable under drought conditions | [53] |

statistical comparisons of the different indices were performed and are presented in Table 2. Drought intensity index (DII) for VDS was 0.71 and significantly lower compared to 0.84, for FDS. In addition, FDS registered significantly higher values for Stress susceptibility percentage index (SSPI = 42.59%) and Percentage of yield reduction (PYR = 83.64%) when compared to VDS (SSPI = 36.42% and PYR = 70.52%, Table 2). The FDS had significantly low values of Yield under stress (Ys = 1.46), Stress tolerance index (STI = 0.18), Geometric mean productivity (GMP = 2.77), Mean productivity (MP = 5.68) and Drought tolerance index (DTI = 0.41) when compared to VDS with Ys = 2.69, STI = 0.33, GMP = 4.32, MP = 6.29 and DTI = 0.51 (Table 2). These results demonstrate that stress induced at the flowering stage is more sensitive

**Table 2. Mean yield under stress and drought tolerance indices for vegetative drought stress (VDS) and flowering drought stress (FDS) for cowpea genotypes.**

| | YS | SSI | RDI | SSPI | SDI | TOL | STI | REI | GMP | MP | YSI | YI | PYR | DTI | HM | DII |
|---|-----|-----|-----|------|-----|-----|-----|-----|-----|----|-----|----|-----|-----|----|-----|
| **VDS** | 2.69[a] | 0.93[a] | 1.09[a] | 36.42[b] | 0.71[b] | 7.21[b] | 0.33[a] | 1.23[a] | 4.32[a] | 6.29[a] | 0.29[a] | 1.01[a] | 70.52[b] | 0.51[a] | 3.66[a] | 0.70[b] |
| **FDS** | 1.47[b] | 0.96[a] | 1.15[a] | 42.59[a] | 0.84[a] | 8.43[a] | 0.18[b] | 1.24[a] | 2.77[b] | 5.68[b] | 0.16[b] | 1.04[a] | 83.64[a] | 0.41[b] | 2.16[b] | 0.84[a] |

YS = Mean value of grain yield under drought stress condition, SSI = Stress susceptibility index, RDI = relative drought index, SSPI = Stress susceptibility percentage index, SDI = Sensitivity drought index, TOL = stress tolerance, STI = Stress tolerance index, REI = The relative efficiency index, GMP = Geometric mean productivity, MP = Mean productivity, YSI = Yield stability index, YI = Yield index, PYR = percentage of yield reduction, DTI = Drought tolerance index, HM = Harmonic mean, DII = Drought Intensity Index, VDS = Vegetative drought stress, FDS = Flowering drought stress. In the same column, means followed by the same letter are not significantly different at 0.050 probability level

and impacted negatively cowpea's productivity compared to the stress that is activated during vegetative stage

## Drought tolerance indices and ranking of cowpea genotypes

Under VDS genotypes C-9, CP-016, C-10, CP-014, C-7gave the highest STI (3.66, 2.88, 1.17, 1.13 and 1.06 respectively), GMP (18.92, 16.78, 10.70, 10.54 and 10.22 respectively) and REI (13.58, 10.68, 4.34, 4.22 and 3.96 respectively) values. Considering FDS, theses indices were high for CP-016 (STI = 2.82; GMP = 16.63 and REI = 19.93), CP-042 (STI = 1.13; GMP = 10.51 and REI = 7.91), CP-056 (STI = 0.85; GMP = 9.12 and REI = 5.96), C-9 (STI = 0.77; GMP = 8.70 and REI = 5.43) and CP-021 (STI = 063; GMP = 7.74 and REI = 4.41). These high indices values are indicative of high yielding genotypes under both drought stress and non-stress conditions (Tables 3 and 4). The lowest STI, GMP and REI values (zero) were obtained by many genotypes under drought stress at both stages, signifying their sensitivity to drought. Under FDS, CP-016, C-9 and MTA-22displayed highest YI (5.65, 3.15 and 2.91 respectively), HM (16.88, 14.03 and 8.10 respectively) and grain yield (15.05, 8.40 and 7.76 g/plant respectively). However, several genotypes showed the lowest estimate for these indices during the VDS (Table 3). With regards to FDS, genotypes CP-016, CP-056 and CP-021displayed highest YI (respectively 10.48, 5.79 and 4.21), HM (respectively 16.51, 9.06 and 7.54) and grain yield (respectively 14.77, 8.16 and 5.94 g/plant) while quite a number of genotypes showed the lowest values that are zero (Table 4). Based on YSI and RDI indices, genotypes CP-046 (YSI = 0.95; RDI = 3.53), MTA-22 (YSI = 0.91; RDI = 3.39) and CP-060 (YSI = 0.91; RDI = 3.38) were found with highest values while many genotypes displayed the lowest amount (zero) for these indices during VDS (Table 3). The SSPI and TOL indices for VDS (Table 3) were low for CP-046 (SSPI = 1.92; TOL = 0.38), MTA-22 (SSPI = 3.64; TOL = 0.72) and CP-060 (SSPI = 3.65; TOL = 0.73) and high for C-9 (SSPI = 172.95; TOL = 34.23). These indices were low for NDT-011, CP-049 and CP-025 and high for C-9, CP-004 and C-7 during FDS (Table 4). During VDS and regarding SSI, SDI and PYR indices, CP-046 (SSI = 0.06; SDI = 0.05 and PYR = 4.88%), MTA-22 (SSI = 0.11; SDI = 0.08 and PYR = 8.49%), CP-060 (SSI = 0.11; SDI = 0.09 and PYR = 8.75%) and CP-016 (SSI = 0.25; SDI = 0.19 and PYR = 19.65%) had low values and many genotypes had maximum values: SSI = 1.30; SDI = 1.31 and PYR = 100%) and regarding FDS, lowest values for SSI, SDI and PYR were for CP-056 (SSI = 0.23; SDI = 0.20 and PYR = 19.94%), CP-016 (SSI = 0.24; SDI = 0.21 and PYR = 21.07%) and CP-049 (SSI = 0.30; SDI = 0.26 and PYR = 26.25%) and the high values recorded for many genotypes with SSI = 1.14; SDI = 1.00 and PYR = 100% (Tables 3 and 4). Drought tolerance index (DTI) at VDS had high values for CP-016 (4.54), MTA-22 (2.67), CP-046 (2.65), CP-060 (2.60) and CP-021 (1.90) and the lowest amount (zero) were for many genotypes (Table 3). At the FDS, higher DTI values were for CP-016 (8.28), CP-056 (4.64), CP-021(2.41), CP-049 (2.28) and NDT-006 (1.36) and lowest values (zero) recorded for many genotypes (Table 4). Taking into consideration all indices under VDS, the top five genotypes named CP 016, CP 021, MTA22, CP 056 and CP060 exhibited the best sum of ranks, respectively of 14.9, 18.67, 19.87, 20.91 and 21.22 and hence, they were identified as the most tolerant genotypes for drought stress induced at the vegetative stage (Table 3). For stress activated at flowering stage (FDS), the top five most drought-tolerant genotypes according to rank sum were CP-016 (7.00), CP-056 (12.33), CP-021 (17.35), CP-028 (28.36) and MTA-22 (30.07) (Table 4). However, genotypes namely CP-037 (RS = 85.47), NDT-011 (RS = 84.72), CP-036 (RS = 83.84), NDT-001(RS = 82.98) and NDT-002 (RS = 81.11) were the top five most drought susceptible genotypes for stress induced at the vegetative stage (Table 3) while the top five susceptible genotypes for stress at the flowering stage were CP-037 (RS = 72.26), CP-034 (RS = 71.34), NDT-001 (RS = 70.80, CP-036 (RS = 70.01) and CP-031(RS = 69.29) (Table 4).

**Table 3. Mean grain yield (g/plant) under VDS (Ysi$_1$) and NDS (Ypi) conditions as well as measures of different drought tolerance indices, rank mean, SD of rank mean and rank sum for 88 cowpea genotypes.**

| Genotype | Ypi | Ysi$_1$ | SSI | RDI | SSPI | SDI | TOL | STI | REI | GMP | MP | YSI | YI | PYR | DTI | HM | DII | Rank Mean | SD of Rank | Rank Sum |
|---|---|---|---|---|---|---|---|---|---|---|---|---|---|---|---|---|---|---|---|---|
| CP-070 | 9.72 (35) | 3.19 (33) | 0.88 (38) | 1.22 (39) | 32.98 (52) | 0.67 (39) | 6.53 (51) | 0.316 (29) | 1.18 (29) | 5.57 (29) | 6.45 (35) | 0.336 (15) | 1.20 (33) | 67.16 (39) | 0.39 (36) | 4.81 (32) | 0.672 (39) | 36.59 | 6.79 | 43.38 |
| CP-071 | 8.992 (42) | 1.35 (55) | 1.11 (52) | 0.56 (52) | 38.63 (55) | 0.85 (52) | 7.65 (54) | 0.123 (50) | 0.45 (50) | 3.47 (50) | 5.17 (53) | 0.244 (25) | 0.50 (55) | 85.03 (52) | 0.075 (55) | 2.34 (51) | 0.85 (52) | 51.71 | 3.37 | 55.07 |
| CP-044 | 10.98 (28) | 3.78 (28) | 0.86 (35) | 1.28 (36) | 36.41 (54) | 0.66 (36) | 7.21 (53) | 0.423 (23) | 1.57 (23) | 6.44 (23) | 7.38 (29) | 0.200 (32) | 1.41 (28) | 65.62 (36) | 0.49 (30) | 5.62 (21) | 0.66 (36) | 32.71 | 9.60 | 42.30 |
| CP-042 | 19.76 (4) | 3.22 (32) | 1.09 (50) | 0.61 (50) | 83.57 (83) | 0.84 (50) | 16.54 (81) | 0.649 (14) | 2.41 (14) | 7.97 (14) | 11.49 (6) | 0.282 (22) | 1.20 (32) | 83.71 (50) | 0.20 (46) | 5.54 (23) | 0.84 (50) | 38.29 | 23.96 | 62.25 |
| CP-018 | 15.20 (13) | 5.70 (11) | 0.81 (30) | 1.39 (31) | 47.99 (68) | 0.62 (31) | 9.49 (66) | 0.884 (6) | 3.28 (6) | 9.30(6) | 10.45 (9) | 0.184 (35) | 2.14 (11) | 62.49 (31) | 0.80 (18) | 8.29 (6) | 0.625 (31) | 23.94 | 19.64 | 43.58 |
| CP-043 | 10.50 (30) | 0.00 (69) | 1.30 (69) | 0.00 (69) | 53.03 (71) | 1.00 (69) | 10.49 (69) | 0(69) | 0(69) | 0(69) | 5.25 (52) | 0(53) | 0 (69) | 100 (69) | 0(69) | 0(69) | 1 (69) | 65.94 | 10.18 | 76.12 |
| CP-056 | 10.19 (32) | 6.70 (9) | 0.45 (11) | 2.44 (12) | 17.62 (25) | 0.34 (12) | 3.48 (25) | 0.697 (12) | 2.59 (12) | 8.26 (12) | 8.45 (17) | 0.800 (1) | 2.52 (9) | 34.22 (12) | 1.65 (9) | 8.09 (8) | 0.34 (12) | 14.18 | 6.74 | 20.91 |
| CP-067 | 11.74 (24) | 0.62 (63) | 1.23 (65) | 0.19 (65) | 56.22 (73) | 0.95 (65) | 11.13 (71) | 0.073 (56) | 0.27 (56) | 2.68 (56) | 6.18 (36) | 0.059 (49) | 0.23 (63) | 94.75 (65) | 0.012 (64) | 1.17 (62) | 0.95 (65) | 59.76 | 12.36 | 72.13 |
| CP-002 | 6.24 (65) | 3.78 (27) | 0.51 (14) | 2.25 (15) | 12.41 (16) | 0.39 (15) | 2.45 (16) | 0.24 (37) | 0.89 (37) | 4.85 (37) | 5.012 (56) | 0.306 (17) | 1.42 (27) | 39.36 (15) | 0.86 (16) | 4.71 (33) | 0.39 (15) | 26.82 | 15.56 | 42.38 |
| CP-006 | 8.48 (45) | 2.94 (36) | 0.85 (34) | 1.29 (35) | 27.97 (40) | 0.65 (35) | 5.53 (40) | 0.25 (35) | 0.94 (35) | 4.99 (35) | 5.71 (44) | 0(53) | 1.10 (36) | 65.28 (35) | 0.38 (37) | 4.37 (36) | 0.65 (34) | 36.94 | 3.31 | 40.25 |
| CP-037 | 1.92 (88) | 0.00 (69) | 1.30 (69) | 0.00 (69) | 9.70 (11) | 1.00 (69) | 1.92 (11) | 0.00 (69) | 0(69) | 0(69) | 0.96 (88) | 0(53) | 0 (69) | 100 (69) | 0(69) | 0(69) | 1 (69) | 64.41 | 21.05 | 85.47 |
| CP-035 | 6.24 (66) | 1.63 (49) | 0.96 (45) | 0.97 (46) | 23.28 (32) | 0.74 (46) | 4.61 (32) | 0.104 (54) | 0.38 (54) | 3.19 (54) | 3.93 (65) | 0(53) | 0.61 (49) | 73.85 (46) | 0.16 (48) | 2.58 (49) | 0.74 (46) | 48.65 | 8.87 | 57.51 |
| CP-058 | 9.44 (38) | 4.55 (19) | 0.67 (19) | 1.79 (20) | 24.70 (36) | 0.52 (20) | 4.88 (36) | 0.438 (22) | 1.63 (22) | 6.55 (22) | 6.99 (30) | 0.259 (24) | 1.71 (19) | 51.79 (20) | 0.82 (17) | 6.14 (19) | 0.52 (20) | 23.47 | 6.87 | 30.35 |
| CP-034 | 19.68 (5) | 2.19 (43) | 1.16 (58) | 0.41 (58) | 88.35 (85) | 0.89 (58) | 17.48 (83) | 0.440 (21) | 1.63 (21) | 6.56 (21) | 10.94 (8) | 0(53) | 0.82 (43) | 88.86 (58) | 0.09 (52) | 3.94 (37) | 0.89 (58) | 45.24 | 23.76 | 68.99 |
| CP-009 | 14.77 (17) | 5.32 (13) | 0.83 (32) | 1.34 (33) | 47.69 (66) | 0.64 (33) | 9.44 (64) | 0.803 (7) | 2.98 (7) | 8.87(7) | 10.04 (10) | 0.134 (39) | 1.99 (13) | 63.92 (33) | 0.72 (21) | 7.83 (10) | 0.641 (33) | 25.53 | 18.57 | 44.10 |
| CP-001 | 9.36 (39) | 3.88 (24) | 0.76 (28) | 1.54 (29) | 27.71 (39) | 0.59 (29) | 5.48 (39) | 0.370 (26) | 1.37 (26) | 6.02 (26) | 6.62 (33) | 0(53) | 1.45 (24) | 58.59 (29) | 0.60 (27) | 5.48 (24) | 0.59 (29) | 29.41 | 5.12 | 34.54 |
| CP-017 | 11.34 (26) | 3.51 (30) | 0.90 (41) | 1.15 (42) | 39.53 (56) | 0.69 (42) | 7.82 (55) | 0.406 (24) | 1.51 (24) | 6.30 (24) | 7.42 (28) | 0(53) | 1.31 (30) | 69.02 (42) | 0.41 (34) | 5.36 (25) | 0.69 (42) | 35.76 | 10.60 | 46.36 |
| CP-019 | 2.74 (84) | 1.65 (48) | 0.52 (15) | 2.23 (16) | 5.53 (6) | 0.40 (16) | 1.10 (6) | 0.046 (62) | 0.17 (62) | 2.12 (62) | 2.19 (82) | 0.168 (37) | 0.62 (48) | 39.92 (16) | 0.37 (38) | 2.06 (57) | 0.40 (16) | 38.24 | 26.62 | 64.85 |
| CP-033 | 8.93 (43) | 6.62 (10) | 0.34 (7) | 2.75 (7) | 11.68 (15) | 0.26 (7) | 2.31 (15) | 0.603 (16) | 2.24 (16) | 7.68 (16) | 7.77 (25) | 0(53) | 2.48 (10) | 25.89 (7) | 1.84 (6) | 7.60 (13) | 0.26 (7) | 13.35 | 9.22 | 22.57 |
| CP-045 | 8.08 (53) | 2.60 (40) | 0.88 (39) | 1.20 (40) | 27.69 (38) | 0.68 (40) | 5.48 (38) | 0.214 (39) | 0.80 (39) | 4.58 (39) | 5.34 (50) | 0(53) | 0.97 (40) | 67.82 (40) | 0.31 (41) | 3.94 (38) | 0.68 (40) | 40.82 | 4.14 | 44.97 |
| CP-060 | 8.32 (48) | 7.59 (5) | 0.11 (3) | 3.39 (3) | 3.68 (3) | 0.09 (3) | 0.72 (3) | 0.644 (15) | 2.39 (15) | 7.94 (15) | 7.96 (23) | 0.230 (29) | 2.85 (5) | 8.75 (3) | 2.60 (4) | 7.94 (9) | 0.09 (3) | 9.59 | 11.63 | 21.22 |
| CP-014 | 14.34 (19) | 7.76 (3) | 0.60 (17) | 2.01 (18) | 33.22 (53) | 0.46 (18) | 6.57 (52) | 1.135 (4) | 4.22 (4) | 10.54 (4) | 11.04 (7) | 0.234 (27) | 2.92 (3) | 45.87 (18) | 1.57 (10) | 10.07 (3) | 0.46 (18) | 25.12 | 19.24 | 44.36 |
| CP-041 | 5.59 (74) | 3.82 (26) | 0.41 (10) | 2.54 (11) | 8.93 (10) | 0.32 (11) | 1.76 (10) | 0.218 (38) | 0.81 (38) | 4.62 (38) | 4.71 (59) | 0.194 (33) | 1.43 (26) | 31.61 (11) | 0.98 (14) | 4.54 (34) | 0.32 (11) | 26.31 | 19.24 | 45.55 |
| CP-057 | 11.20 (27) | 4.84 (18) | 0.74 (23) | 1.61 (24) | 32.13 (50) | 0.57 (24) | 6.36 (49) | 0.553 (19) | 2.06 (19) | 7.36 (19) | 8.02 (22) | 0.298 (18) | 1.81 (18) | 56.78 (24) | 0.78 (19) | 6.76 (16) | 0.57 (24) | 24.38 | 10.40 | 34.78 |
| CP-007 | 8.28 (50) | 2.05 (45) | 0.98 (46) | 0.92 (47) | 31.49 (46) | 0.75 (47) | 6.23 (46) | 0.173 (43) | 0.64 (43) | 4.11 (43) | 5.16 (54) | 0.236 (26) | 0.76 (45) | 75.26 (47) | 0.19 (47) | 3.28 (43) | 0.75 (47) | 46.19 | 2.77 | 48.96 |
| CP-030 | 6.88 (57) | 0.91 (58) | 1.13 (54) | 0.49 (54) | 30.15 (44) | 0.87 (54) | 5.96 (44) | 0.064 (58) | 0.25 (58) | 2.50 (58) | 3.89 (66) | 0.193 (34) | 0.34 (58) | 86.74 (54) | 0.046 (57) | 1.61 (59) | 0.87 (54) | 55.44 | 5.22 | 60.66 |

*(Continued)*

**Table 3.** (Continued)

| Genotype | Ypi | Ysi₁ | SSI | RDI | SSPI | SDI | TOL | STI | REI | GMP | MP | YSI | YI | PYR | DTI | HM | DII | Rank Mean | SD of Rank | Rank Sum |
|---|---|---|---|---|---|---|---|---|---|---|---|---|---|---|---|---|---|---|---|---|
| CP-020 | 15.44 (12) | 0.82 (60) | 1.23 (64) | 0.20 (64) | 73.88 (78) | 0.95 (64) | 14.62 (76) | 0.128 (49) | 0.47 (49) | 3.54 (49) | 8.13 (20) | 0.227 (30) | 0.31 (60) | 94.71 (64) | 0.016 (62) | 1.55 (60) | 0.95 (64) | 56.06 | 17.34 | 73.40 |
| CP-013 | 6.04 (70) | 2.78 (37) | 0.70 (20) | 1.71 (21) | 16.49 (24) | 0.54 (21) | 3.26 (24) | 0.171 (44) | 0.63 (44) | 4.09 (44) | 4.40 (63) | 0.284 (21) | 1.03 (37) | 54.04 (21) | 0.47 (32) | 3.80 (40) | 0.54 (21) | 35.19 | 15.28 | 50.46 |
| CP-012 | 14.56 (18) | 5.21 (15) | 0.84 (33) | 1.33 (34) | 47.23 (63) | 0.64 (34) | 9.34 (62) | 0.774 (8) | 2.87 (8) | 8.71(8) | 9.88 (12) | 0(53) | 1.95 (15) | 64.28 (34) | 0.70 (23) | 7.67 (11) | 0.642 (34) | 25.81 | 17.34 | 43.16 |
| CP-046 | 7.80 (55) | 7.42 (6) | 0.06 (1) | 3.53 (1) | 1.92 (1) | 0.05 (1) | 0.38 (1) | 0.591 (17) | 2.19 (17) | 7.61 (17) | 7.62 (26) | 0.367 (14) | 2.78 (6) | 4.89 (1) | 2.65 (3) | 7.61 (12) | 0.05 (1) | 10.38 | 14.05 | 24.43 |
| CP-003 | 9.58 (37) | 0.35 (65) | 1.26 (66) | 0.14 (66) | 46.62 (61) | 0.96 (66) | 9.22 (60) | 0.034 (63) | 0.12 (63) | 1.83 (63) | 4.97 (57) | 0.066 (48) | 0.13 (65) | 96.32 (66) | 0.004 (66) | 0.68 (65) | 0.96 (66) | 62.25 | 7.03 | 69.28 |
| CP-005 | 4.57 (77) | 1.38 (54) | 0.91 (43) | 1.12 (44) | 16.13 (23) | 0.70 (44) | 3.19 (23) | 0.064 (57) | 0.26 (57) | 2.50 (57) | 2.972 (76) | 0(53) | 0.52 (54) | 69.87 (44) | 0.15 (49) | 2.12 (55) | 0.698 (44) | 50.06 | 14.35 | 64.41 |
| CP-004 | 25.04 (2) | 2.98 (35) | 1.15 (56) | 0.44 (56) | 111.43 (87) | 0.88 (56) | 22.05 (85) | 0.762 (11) | 2.83 (11) | 8.6411) | 14.01 (3) | 0.028 (52) | 1.12 (35) | 88.087 (56) | 0.13 (50) | 5.33 (26) | 0.88 (56) | 39.88 | 26.95 | 66.83 |
| CP-054 | 11.52 (25) | 0.25 (67) | 1.28 (67) | 0.08 (67) | 56.95 (74) | 0.98 (67) | 11.27 (72) | 0.029 (65) | 0.11 (65) | 1.69 (65) | 5.88 (42) | 0.291 (20) | 0.09 (67) | 97.84 (67) | 0.002 (67) | 0.48 (67) | 0.98 (67) | 63.31 | 11.97 | 75.29 |
| CP-021 | 10.35 (31) | 7.24 (7) | 0.39 (9) | 2.60 (9) | 15.72 (22) | 0.30 (9) | 3.11 (22) | 0.765 (9) | 2.84 (9) | 8.66(9) | 8.79 (15) | 0.573 (5) | 2.72 (7) | 30.06 (9) | 1.904 (5) | 8.52 (5) | 0.30 (9) | 11.63 | 7.05 | 18.67 |
| CP-027 | 16.22 (10) | 1.10 (57) | 1.21 (60) | 0.25 (60) | 76.40 (81) | 0.93 (60) | 15.12 (79) | 0.182 (41) | 0.68 (41) | 4.22 (41) | 8.66 (16) | 0.207 (31) | 0.41 (57) | 93.21 (60) | 0.028 (59) | 2.06 (56) | 0.932 (60) | 52.50 | 18.69 | 71.19 |
| MTA-22 | 8.48 (46) | 7.76 (4) | 0.11 (2) | 3.40 (2) | 3.64 (2) | 0.08 (2) | 0.72 (2) | 0.671 (13) | 2.49 (13) | 8.11 (13) | 8.12 (21) | 0.330 (16) | 2.91 (4) | 8.49 (2) | 2.66 (2) | 8.10 (7) | 0.08 (2) | 8.56 | 11.30 | 19.87 |
| CP-036 | 16.62 (8) | 0.00 (69) | 1.30 (69) | 0.00 (69) | 83.99 (84) | 1.00 (69) | 16.62 (82) | 0(69) | 0(69) | 0(69) | 8.34 (18) | 0(53) | 0 (69) | 100 (69) | 0(69) | 0(69) | 1 (69) | 63.88 | 19.96 | 83.84 |
| CP-026 | 16.72 (7) | 1.93 (46) | 1.15 (57) | 0.43 (57) | 74.74 (79) | 0.88 (57) | 14.79 (77) | 0.328 (27) | 1.22 (27) | 5.67 (27) | 9.32 (13) | 0.091 (45) | 0.72 (46) | 88.48 (57) | 0.08 (54) | 3.45 (42) | 0.885 (57) | 45.75 | 20.42 | 66.17 |
| CP-025 | 2.21 (86) | 0.70 (61) | 0.89 (40) | 1.18 (41) | 7.62 (9) | 0.68 (41) | 1.51 (9) | 0.015 (67) | 0.06 (67) | 1.24 (67) | 1.45 (85) | 0.297 (19) | 0.26 (61) | 68.29 (41) | 0.084 (53) | 1.06 (63) | 0.683 (41) | 52.31 | 22.06 | 74.37 |
| CP-069 | 8.38 (47) | 2.50 (41) | 0.92 (44) | 1.11 (45) | 29.75 (43) | 0.70 (45) | 5.88 (43) | 0.213 (40) | 0.79 (40) | 4.57 (40) | 5.44 (49) | 0.118 (43) | 0.94 (41) | 70.23 (45) | 0.27 (42) | 3.84 (39) | 0.70 (45) | 43.06 | 2.79 | 45.85 |
| CP-029 | 8.24 (51) | 3.00 (34) | 0.83 (31) | 1.35 (32) | 26.47 (37) | 0.64 (32) | 5.24 (37) | 0.252 (36) | 0.93 (36) | 4.97 (36) | 5.62 (47) | 0(53) | 1.13 (34) | 63.59 (32) | 0.42 (33) | 4.39 (35) | 0.64 (32) | 35.94 | 5.42 | 41.35 |
| CP-049 | 5.91 (71) | 0.35 (66) | 1.23 (62) | 0.22 (62) | 28.09 (41) | 0.94 (62) | 5.56 (41) | 0.021 (66) | 0.08 (66) | 1.44 (66) | 3.132 (73) | 0.737 (3) | 0.13 (66) | 94.04 (62) | 0.008 (65) | 0.66 (6) | 0.94 (62) | 62.50 | 8.64 | 71.14 |
| CP-008 | 6.40 (62) | 0.00 (69) | 1.30 (69) | 0.00 (69) | 32.33 (51) | 1.00 (69) | 6.4 (50) | 0(69) | 0(69) | 0(69) | 3.20 (70) | 0.281 (24) | 0 (69) | 100 (69) | 0(69) | 0(69) | 1 (69) | 66.38 | 6.09 | 72.47 |
| CP-048 | 6.51 (61) | 4.04 (23) | 0.49 (13) | 2.30 (14) | 12.49 (17) | 0.38 (14) | 2.47 (17) | 0.268 (34) | 0.99 (34) | 5.13 (34) | 5.28 (51) | 0.463 (7) | 1.52 (23) | 37.97 (14) | 0.94 (15) | 4.98 (30) | 0.38 (14) | 25.50 | 14.20 | 39.70 |
| CP-068 | 6.72 (59) | 5.57 (12) | 0.22 (5) | 3.08 (5) | 5.82 (7) | 0.17 (5) | 1.15 (7) | 0.382 (25) | 1.42 (25) | 6.11 (25) | 6.14 (38) | 0.441 (9) | 2.09 (12) | 17.14 (5) | 1.73 (7) | 6.09 (20) | 0.17 (5) | 16.38 | 14.98 | 31.36 |
| CP-016 | 18.72 (6) | 15.05 (1) | 0.26 (6) | 2.99 (6) | 18.55 (28) | 0.20 (6) | 3.67 (28) | 2.876 (2) | 10.68 (2) | 16.78 (2) | 16.88 (2) | 0.789 (2) | 5.64 (1) | 19.61 (6) | 4.54 (1) | 16.68 (1) | 0.20 (6) | 6.50 | 8.40 | 14.90 |
| CP-031 | 3.26 (82) | 2.32 (42) | 0.38 (8) | 2.64 (8) | 4.77 (5) | 0.29 (8) | 0.94 (5) | 0.077 (55) | 0.28 (55) | 2.75 (55) | 2.79 (78) | 0(53) | 0.87 (42) | 28.92 (8) | 0.618 (26) | 2.71 (48) | 0.29 (8) | 23.94 | 24.74 | 48.68 |
| CP-028 | 8.30 (49) | 3.60 (29) | 0.74 (22) | 1.61 (23) | 23.78 (33) | 0.57 (23) | 4.70 (33) | 0.305 (30) | 1.13 (30) | 5.46 (30) | 5.95 (40) | 0.390 (12) | 1.35 (29) | 56.68 (23) | 0.58 (28) | 5.02 (29) | 0.57 (23) | 29.63 | 7.00 | 36.63 |
| CP-024 | 8.24 (52) | 3.49 (31) | 0.75 (25) | 1.57 (26) | 24.01 (34) | 0.58 (26) | 4.75 (34) | 0.293 (33) | 1.08 (33) | 5.36 (33) | 5.86 (43) | 0.078 (47) | 1.31 (31) | 57.67 (26) | 0.55 (29) | 4.90 (31) | 0.58 (26) | 32.06 | 6.94 | 39.01 |
| C | 4.00 (78) | 2.77 (38) | 1.05 (48) | 2.57 (10) | 6.22 (8) | 0.31 (10) | 1.23 (8) | 0.113 (51) | 0.42 (51) | 3.32 (51) | 3.38 (69) | 0(53) | 1.04 (38) | 30.8 (10) | 0.71 (22) | 3.27 (44) | 0.31 (10) | 34.13 | 23.32 | 57.45 |
| D | 6.07 (69) | 5.22 (14) | 0.18 (4) | 3.20 (4) | 4.28 (4) | 0.14 (4) | 0.85 (4) | 0.323 (28) | 1.21 (28) | 5.63 (28) | 5.65 (46) | 0(53) | 1.96 (14) | 13.97 (4) | 1.68 (8) | 5.62 (22) | 0.14 (4) | 17.81 | 18.34 | 36.16 |

(*Continued*)

**Table 3.** (Continued)

| Genotype | Ypi | Ysi₁ | SSI | RDI | SSPI | SDI | TOL | STI | REI | GMP | MP | YSI | YI | PYR | DTI | HM | DII | Rank Mean | SD of Rank | Rank Sum |
|---|---|---|---|---|---|---|---|---|---|---|---|---|---|---|---|---|---|---|---|---|
| NDT-003 | 3.81 (79) | 1.61 (50) | 0.75 (26) | 1.57 (27) | 11.11 (14) | 0.58 (27) | 2.2 (14) | 0.062 (60) | 0.23 (60) | 2.47 (60) | 2.71 (79) | 0(53) | 0.60 (50) | 57.77 (27) | 0.25 (43) | 2.26 (52) | 0.58 (27) | 43.44 | 20.74 | 64.18 |
| F | 6.22 (68) | 0.00 (69) | 1.30 (69) | 0.00 (69) | 31.45 (45) | 1.00 (69) | 6.22 (45) | 0(69) | 0(69) | 0(69) | 3.11 (75) | 0(53) | 0 (69) | 100 (69) | 0(69) | 0(69) | 1 (69) | 66.31 | 8.22 | 74.53 |
| B | 6.24 (67) | 0.00 (69) | 1.30 (69) | 0.00 (69) | 31.53 (47) | 1.00 (69) | 6.24 (47) | 0(69) | 0(69) | 0(69) | 3.12 (74) | 0(53) | 0 (69) | 100 (69) | 0(69) | 0(69) | 1 (69) | 66.44 | 7.49 | 73.93 |
| NDT-010 | 14.98 (14) | 5.00 (17) | 0.87 (37) | 1.24 (38) | 50.40 (70) | 0.67 (38) | 9.97 (68) | 0.764 (10) | 2.84 (10) | 8.65 (10) | 9.98 (11) | 0.233 (28) | 1.87 (17) | 66.63 (38) | 0.62 (24) | 7.49 (15) | 0.67 (38) | 28.56 | 19.32 | 47.88 |
| NDT-016 | 3.62 (80) | 1.50 (52) | 0.76 (27) | 1.54 (28) | 10.69 (13) | 0.59 (28) | 2.11 (13) | 0.055 (61) | 0.21 (61) | 2.32 (61) | 2.56 (80) | 0.446 (8) | 0.56 (52) | 58.51 (28) | 0.23 (45) | 2.12 (54) | 0.59 (28) | 44.44 | 21.18 | 65.61 |
| NDT-045 | 6.80 (58) | 4.29 (21) | 0.48 (12) | 2.34 (13) | 12.69 (18) | 0.37 (13) | 2.51 (18) | 0.297 (31) | 1.11 (31) | 5.39 (31) | 5.54 (48) | 0.168 (38) | 1.61 (21) | 36.94 (13) | 1.01 (13) | 5.26 (27) | 0.37 (13) | 23.81 | 13.31 | 37.13 |
| NDT-004 | 10.90 (29) | 1.42 (53) | 1.13 (55) | 0.48 (55) | 47.90 (67) | 0.87 (55) | 9.48 (65) | 0.157 (46) | 0.58 (46) | 3.92 (46) | 6.16 (37) | 0.127 (42) | 0.53 (53) | 87.01 (55) | 0.069 (56) | 2.51 (50) | 0.87 (55) | 51.56 | 9.31 | 60.87 |
| NDT-002 | 14.88 (16) | 0.00 (69) | 1.30 (69) | 0.00 (69) | 75.18 (80) | 1.00 (69) | 14.88 (78) | 0(69) | 0(69) | 0(69) | 7.44 (27) | 0(53) | 0 (69) | 100 (69) | 0(69) | 0(69) | 1 (69) | 64.44 | 16.77 | 81.21 |
| E | 14.98 (15) | 0.86 (59) | 1.23 (63) | 0.21 (63) | 71.30 (77) | 0.94 (63) | 14.11 (75) | 0.132 (48) | 0.49 (48) | 3.597 (48) | 7.92 (24) | 0(53) | 0.33 (59) | 94.23 (63) | 0.019 (61) | 1.63 (58) | 0.942 (63) | 55.56 | 16.05 | 71.61 |
| C-9 | 42.63 (1) | 8.40 (2) | 1.05 (47) | 0.73 (48) | 172.95 (88) | 0.80 (48) | 34.23 (86) | 3.656 (1) | 13.59 (1) | 18.92 (1) | 25.52 (1) | 0.041 (51) | 3.15 (2) | 80.29 (48) | 0.619 (25) | 14.03 (2) | 0.80 (48) | 28.19 | 30.83 | 59.02 |
| C-7 | 23.38 (3) | 4.47 (20) | 1.05 (49) | 0.71 (49) | 95.51 (86) | 0.81 (49) | 18.90 (84) | 1.067 (5) | 3.96 (5) | 10.22 (5) | 13.93 (4) | 0.045 (50) | 1.68 (20) | 80.86 (49) | 0.32 (40) | 7.51 (14) | 0.81 (49) | 33.31 | 27.35 | 60.66 |
| NDT-029 | 3.20 (83) | 0.44 (64) | 1.12 (53) | 0.51 (53) | 13.94 (20) | 0.86 (53) | 2.76 (20) | 0.014 (68) | 0.05 (68) | 1.18 (68) | 1.82 (83) | 0.407 (11) | 0.16 (64) | 86.25 (53) | 0.023 (60) | 0.77 (64) | 0.86 (53) | 57.94 | 17.14 | 75.07 |
| C-14 | 10.18 (33) | 1.60 (51) | 1.10 (51) | 0.58 (51) | 43.37 (58) | 0.84 (51) | 8.58 (57) | 0.166 (45) | 0.62 (45) | 4.03 (45) | 5.89 (41) | 0(53) | 0.60 (51) | 84.28 (51) | 0.1 (51) | 2.76 (47) | 0.84 (51) | 48.75 | 5.97 | 54.72 |
| C-10 | 16.36 (9) | 7.00 (8) | 0.75 (24) | 1.59 (25) | 47.2 (64) | 0.57 (25) | 9.36 (63) | 1.169 (3) | 4.34 (3) | 10.70 (3) | 11.68 (5) | 0.104 (44) | 2.62 (8) | 57.21 (25) | 1.12 (11) | 9.80 (4) | 0.572 (25) | 19.13 | 19.12 | 38.24 |
| C-18 | 13.52 (20) | 4.17 (22) | 0.90 (42) | 1.15 (43) | 47.22 (62) | 0.69 (43) | 9.34 (61) | 0.576 (18) | 2.14 (18) | 7.51 (18) | 8.84 (14) | 0(53) | 1.56 (22) | 69.13 (43) | 0.48 (31) | 6.38 (18) | 0.691 (43) | 32.44 | 15.69 | 48.13 |
| C-19 | 9.60 (36) | 0.64 (62) | 1.22 (61) | 0.25 (61) | 45.27 (59) | 0.93 (61) | 8.96 (58) | 0.062 (59) | 0.24 (59) | 2.47 (59) | 5.12 (55) | 0(53) | 0.24 (62) | 93.33 (61) | 0.016 (63) | 1.2 (61) | 0.933 (61) | 59.13 | 6.60 | 65.73 |
| NDT-015 | 13.28 (21) | 0.00 (69) | 1.30 (69) | 0.00 (69) | 67.09 (76) | 1.00 (69) | 13.28 (74) | 0(69) | 0(69) | 0(69) | 6.64 (32) | 0(53) | 0 (69) | 100 (69) | 0(69) | 0(69) | 1 (69) | 64.56 | 14.74 | 79.30 |
| NDP-017 | 9.36 (40) | 0.00 (69) | 1.30 (69) | 0.00 (69) | 47.29 (65) | 1.00 (69) | 9.36 (63) | 0(69) | 0(69) | 0(69) | 4.68 (60) | 0(53) | 0 (69) | 100 (69) | 0(69) | 0(69) | 1 (69) | 66.13 | 7.20 | 73.33 |
| NDT-005 | 9.92 (34) | 0.00 (69) | 1.30 (69) | 0.00 (69) | 50.12 (69) | 1.00 (69) | 9.92 (67) | 0(69) | 0(69) | 0(69) | 4.96 (58) | 0(53) | 0 (69) | 100 (69) | 0(69) | 0(69) | 1 (69) | 66.13 | 8.74 | 74.86 |
| NDT-006 | 5.61 (73) | 1.90 (47) | 0.86 (36) | 1.26 (37) | 18.73 (29) | 0.66 (37) | 3.71 (29) | 0.108 (53) | 0.40 (53) | 3.26 (53) | 3.75 (67) | 0.586 (4) | 0.71 (47) | 66.12 (37) | 0.24 (44) | 2.84 (46) | 0.66 (37) | 45.31 | 12.20 | 57.52 |
| NDT-025 | 8.80 (44) | 5.04 (16) | 0.56 (16) | 2.13 (17) | 19.00 (30) | 0.43 (17) | 3.76 (30) | 0.452 (20) | 1.68 (20) | 6.65 (20) | 6.92 (31) | 0(53) | 1.89 (16) | 42.72 (17) | 1.08 (12) | 6.41 (17) | 0.43 (17) | 21.25 | 8.09 | 29.34 |
| NDT-001 | 2.56 (85) | 0.00 (69) | 1.30 (69) | 0.00 (69) | 12.93 (19) | 1.00 (69) | 2.56 (19) | 0(69) | 0(69) | 0(69) | 1.28 (86) | 0(53) | 0 (69) | 100 (69) | 0(69) | 0(69) | 1 (69) | 64.81 | 18.17 | 82.98 |
| A | 7.50 (56) | 3.84 (25) | 0.64 (18) | 1.90 (19) | 18.51 (27) | 0.49 (19) | 3.66 (27) | 0.295 (32) | 1.09 (32) | 5.37 (32) | 5.67 (45) | 0.133 (40) | 1.44 (25) | 48.82 (19) | 0.73 (20) | 5.08 (28) | 0.49 (19) | 27.69 | 10.30 | 37.99 |
| NDT-009 | 12.21 (22) | 0.00 (69) | 1.30 (69) | 0.00 (69) | 61.68 (75) | 1.00 (69) | 12.21 (73) | 0(69) | 0(69) | 0(69) | 6.10 (39) | 0(53) | 0 (69) | 100 (69) | 0(69) | 0(69) | 1 (69) | 64.94 | 13.54 | 78.48 |
| NDT-027 | 4.96 (75) | 2.19 (44) | 0.73 (21) | 1.64 (22) | 13.98 (21) | 0.56 (22) | 2.76 (21) | 0.111 (52) | 0.41 (52) | 3.29 (52) | 3.57 (68) | 0.129 (41) | 0.82 (44) | 55.80 (22) | 0.36 (39) | 3.04 (45) | 0.56 (22) | 38.88 | 17.90 | 56.77 |
| NDT-011 | 2.08 (87) | 0.00 (69) | 1.30 (69) | 0.00 (69) | 10.51 (12) | 1.00 (69) | 2.08 (12) | 0(69) | 0(69) | 0(69) | 1.04 (87) | 0.41 (10) | 0 (69) | 100 (69) | 0(69) | 0(69) | 1 (69) | 64.13 | 20.60 | 84.72 |

(Continued)

**Table 3.** (Continued)

| Genotype | Ypi | Ysi₁ | SSI | RDI | SSPI | SDI | TOL | STI | REI | GMP | MP | YSI | YI | PYR | DTI | HM | DII | Rank Mean | SD of Rank | Rank Sum |
|---|---|---|---|---|---|---|---|---|---|---|---|---|---|---|---|---|---|---|---|---|
| NDT-008 | 11.98 (23) | 1.20 (56) | 1.17 (59) | 0.37 (59) | 54.48 (72) | 0.90 (59) | 10.78 (70) | 0.146 (47) | 0.54 (47) | 3.79 (47) | 6.59 (34) | 0.169 (36) | 0.45 (56) | 89.98 (59) | 0.045 (58) | 2.18 (53) | 0.90 (59) | 53.75 | 12.07 | 65.82 |
| C-8 | 6.59 (60) | 2.67 (39) | 0.77 (29) | 1.51 (30) | 19.80 (31) | 0.59 (30) | 3.92 (31) | 0.179 (42) | 0.67 (42) | 4.19 (42) | 4.63 (61) | 0(53) | 1.00 (39) | 59.46 (30) | 0.40 (35) | 3.80 (41) | 0.594 (30) | 38.25 | 9.93 | 48.18 |
| NDT-013 | 16.16 (11) | 0.19 (68) | 1.29 (68) | 0.04 (68) | 80.67 (82) | 0.99 (68) | 15.96 (80) | 0.031 (64) | 0.12 (64) | 1.76 (64) | 8.17 (19) | 0(53) | 0.07 (68) | 98.81 (68) | 0.001 (68) | 0.38 (68) | 0.99 (68) | 62.38 | 18.73 | 81.11 |
| NDT-022 | 3.54 (81) | 0.00 (69) | 1.30 (69) | 0.00 (69) | 17.86 (26) | 1.00 (69) | 3.53 (26) | 0(69) | 0(69) | 0(69) | 1.76 (84) | 0.520 (6) | 0 (69) | 100 (69) | 0(69) | 0(69) | 1 (69) | 65.31 | 15.54 | 80.85 |
| NDT-018 | 4.80 (76) | 0.00 (69) | 1.30 (69) | 0.00 (69) | 24.25 (35) | 1.00 (69) | 4.8 (35) | 0(69) | 0(69) | 0(69) | 2.40 (81) | 0.368 (13) | 0 (69) | 100 (69) | 0(69) | 0(69) | 1 (69) | 65.94 | 12.15 | 78.09 |
| NDT-026 | 9.10 (41) | 0.00 (69) | 1.30 (69) | 0.00 (69) | 46.00 (60) | 1.00 (69) | 9.1& (59) | 0(69) | 0(69) | 0(69) | 4.55 (62) | 0(53) | 0 (69) | 100 (69) | 0(69) | 0(69) | 1 (69) | 65.69 | 7.20 | 72.89 |
| NDT-020 | 6.32 (63) | 0.00 (69) | 1.30 (69) | 0.00 (69) | 31.93 (48) | 1.00 (69) | 6.32 (48) | 0(69) | 0(69) | 0(69) | 3.16 (71) | 0.079 (46) | 0 (69) | 100 (69) | 0(69) | 0(69) | 1 (69) | 66.13 | 7.06 | 73.19 |
| NDT-023 | 7.89 (54) | 0.00 (69) | 1.30 (69) | 0.00 (69) | 39.85 (57) | 1.00 (69) | 7.88 (56) | 0(69) | 0(69) | 0(69) | 3.94 (64) | 0(53) | 0 (69) | 100 (69) | 0(69) | 0(69) | 1 (69) | 66.25 | 5.15 | 71.40 |
| NDT-012 | 5.79 (72) | 0.00 (69) | 1.30 (69) | 0.00 (69) | 29.26 (42) | 1.00 (69) | 5.79 (42) | 0(69) | 0(69) | 0(69) | 2.89 (77) | 0(53) | 0 (69) | 100 (69) | 0(69) | 0(69) | 1 (69) | 66.31 | 9.43 | 75.74 |
| NDT-025 B | 6.32 (64) | 0.00 (69) | 1.30 (69) | 0.00 (69) | 31.93 (49) | 1.00 (69) | 6.32 (48) | 0(69) | 0(69) | 0(69) | 3.16 (72) | 0(53) | 0 (69) | 100 (69) | 0(69) | 0(69) | 1 (69) | 66.38 | 6.76 | 73.13 |

Values in the parentheses refer to the ranks of the genotype for each index, YSi1 = Mean value of grain yield under drought stress condition, Ypi = mean value of yield under non-stress condition, SSI = Stress susceptibility index, RDI = relative drought index, SSPI = Stress susceptibility percentage index, SDI = Sensitivity drought index, TOL = stress tolerance, STI = Stress tolerance index, REI = The relative efficiency index, GMP = Geometric mean productivity, MP = Mean productivity, YSI = Yield stability index, YI = Yield index, PYR = percentage of yield reduction, DTI = Drought tolerance index, HM = Harmonic mean, DII = Drought Intensity Index

## Correlations between drought tolerant indices and seed yield

To determine the most desirable drought tolerant selection criteria, Pearson correlation plot between Yp, Ys, and the fifteen different indices related to drought tolerance were estimated and are presented in Fig 1. Under both stress at the vegetative (VDS) and flowering stage (FDS), there were positive and significant correlation between Yp and Ys: r = 0.36 and 0.22 for VDS and FDS respectively. Grain yield under both non-stress and stress conditions significantly and positively correlated with STI ($r_p$ = 0.71; $r_s$ = 0.80), REI ($r_p$ = 0.71, $r_s$ = 0.80), GMP ($r_p$ = 0.63; $r_s$ = 0.91), MP ($r_p$ = 0.94; $r_s$ = 0.66), YI ($r_p$ = 0.36; $r_s$ = 1.0) and HM ($r_p$ = 0.49; $r_s$ = 0.98) at VDS (Fig 1A). The same observations were found under FDS STI ($r_p$ = 0.48; $r_s$ = 0.92), REI ($r_p$ = 0.43, $r_s$ = 0.92), GMP ($r_p$ = 0.44; $r_s$ = 0.91), MP ($r_p$ = 0.95; $r_s$ = 0.51), YI ($r_p$ = 0.22; $r_s$ = 1.0) and HM ($r_p$ = 0.27; $r_s$ = 0.97) (Fig 1B). With regards to VDS mean grain yields under stresses conditions (Ys) were significantly and positively correlated with RDI (r = 0.78), STI (r = 0.80), REI (r = 0.80), GMP r = 0.91), MP (r = 0.66), YSI (r = 0.78), YI (r = 1.00), DTI (r = 0.90), and HM (r = 0.98); significantly and negatively associated with these four indices namely SSI (r = -0.79), SDI (-0.78), PYR (r = -0.80) and DII (r = -0.78) (Fig 1A). The same trend was observed under FDS as shown in Fig 1B.

## Principal component analysis

Principal component analysis (PCA) revealed that under VDS, the first 2 principal components, PC1 and PC2 explained respectively 61.44 and 33.13% (both 94.57%) of the total

**Table 4. Mean grain yield (g/plant) under FDS (Ysi$_2$) and NDS (Ypi) conditions as well as measures of different drought tolerance indices, rank mean, SD of rank mean and rank sum for 88 cowpea genotypes.**

| Genotype | Ypi | Ysi$_2$ | SSI | RDI | SSPI | SDI | TOL | STI | REI | GMP | MP | YSI | YI | PYR | DTI | HM | DII | Rank Mean | SD of Rank | Rank Sum |
|---|---|---|---|---|---|---|---|---|---|---|---|---|---|---|---|---|---|---|---|---|
| CP-070 | 9.72 (35) | 3.27 (13) | 0.992 (40) | 5.622 (1) | 32.59 (40) | 0.663 (15) | 6.45 (40) | 0.32 (13) | 2.27 (13) | 5.63 (13) | 6.49 (28) | 0.33 (39) | 2.319 (13) | 66.37 (15) | 0.78 (10) | 4.89 (12) | 0.66 (15) | 20.29 | 11.24 | 31.53 |
| CP-071 | 8.992 (42) | 2.2 (21) | 0.989 (38) | 5.543 (2) | 34.31 (43) | 0.755 (25) | 6.79 (42) | 0.201 (26) | 1.41 (26) | 4.44 (26) | 5.59 (36) | 0.15 (52) | 1.561 (21) | 75.53 (25) | 0.38 (24) | 3.53 (22) | 0.76 (25) | 29.00 | 7.91 | 36.91 |
| CP-044 | 10.98 (28) | 2.2 (22) | 0.913 (29) | 5.179 (3) | 44.37 (53) | 0.799 (32) | 8.78 (52) | 0.246 (21) | 1.72 (21) | 4.91 (21) | 6.59 (27) | 0.34 (36) | 1.561 (22) | 79.97 (32) | 0.30 (31) | 3.66 (21) | 0.80 (32) | 29.94 | 9.83 | 39.77 |
| CP-042 | 19.76 (4) | 5.58 (4) | 0.819 (20) | 4.117 (4) | 71.61 (77) | 0.717 (22) | 14.17 (76) | 1.12 (2) | 7.91 (2) | 10.50 (2) | 12.67 (4) | 0.17 (50) | 3.961 (4) | 71.73 (22) | 1.12 (6) | 8.70 (3) | 0.72 (22) | 18.53 | 23.67 | 42.20 |
| CP-018 | 15.20 (13) | 2.8 (18) | 0.931 (32) | 4.027 (5) | 62.64 (72) | 0.815 (35) | 12.40 (71) | 0.43 (10) | 3.05 (9) | 6.52 (10) | 9.00 (13) | 0.38 (31) | 1.986 (18) | 81.57 (35) | 0.36 (26) | 4.72 (13) | 0.82 (35) | 28.35 | 19.34 | 47.69 |
| CP-043 | 10.50 (30) | 0 (53) | 1.14 (53) | 3.654 (6) | 53.02 (65) | 1 (53) | 10.49 (64) | 0 (53) | 0 (53) | 0 (53) | 5.24 (38) | 0 (69) | 0 (53) | 100 (53) | 0 (53) | 0 (53) | 1 (53) | 52.00 | 8.38 | 60.38 |
| CP-056 | 10.19 (32) | 8.16 (2) | 0.22 (1) | 3.252 (7) | 10.26 (8) | 0.199 (1) | 2.03 (8) | 0.84 (3) | 5.96 (3) | 9.11 (3) | 9.17 (10) | 0.65 (12) | 5.790 (2) | 19.93 (1) | 4.63 (2) | 9.06 (2) | 0.20 (1) | 4.76 | 7.56 | 12.33 |
| CP-067 | 11.74 (24) | 0.704 (46) | 1.074 (49) | 3.138 (8) | 55.77 (67) | 0.940 (49) | 11.04 (66) | 0.084 (39) | 0.58 (39) | 2.87 (39) | 6.22 (29) | 0.05 (65) | 0.499 (46) | 94.00 (49) | 0.029 (50) | 1.32 (45) | 0.94 (49) | 46.18 | 10.82 | 56.99 |
| CP-002 | 6.24 (65) | 1.912 (27) | 0.791 (15) | 3.101 (9) | 21.86 (21) | 0.693 (17) | 4.31 (21) | 0.121 (33) | 0.85 (33) | 3.45 (33) | 4.07 (61) | 0.60 (15) | 1.356 (27) | 69.35 (17) | 0.41 (23) | 2.92 (29) | 0.69 (17) | 27.82 | 14.65 | 42.47 |
| CP-006 | 8.48 (45) | 0 (53) | 1.14 (53) | 2.879 (10) | 42.84 (52) | 1 (53) | 8.48 (51) | 0 (53) | 0 (53) | 0 (53) | 4.24 (57) | 0.35 (35) | 0 (53) | 100 (53) | 0 (53) | 0 (53) | 1 (53) | 52.65 | 2.23 | 54.88 |
| CP-037 | 1.92 (88) | 0 (53) | 1.14 (53) | 2.861 (11) | 9.70 (6) | 1 (53) | 1.92 (6) | 0 (53) | 0 (53) | 0 (53) | 0.96 (88) | 0 (69) | 0 (53) | 100 (53) | 0 (53) | 0 (53) | 1 (53) | 51.59 | 20.67 | 72.26 |
| CP-035 | 6.24 (66) | 0 (53) | 1.14 (53) | 2.740 (12) | 31.52 (35) | 1 (53) | 6.24 (35) | 0 (53) | 0 (53) | 0 (53) | 3.15 (71) | 0.26 (46) | 0 (53) | 100 (53) | 0 (53) | 0 (53) | 1 (53) | 52.71 | 8.44 | 61.15 |
| CP-058 | 9.44 (38) | 2.448 (20) | 0.845 (22) | 2.586 (13) | 35.32 (44) | 0.740 (24) | 6.99 (43) | 0.235 (23) | 1.65 (23) | 4.80 (23) | 5.94 (32) | 0.48 (20) | 1.737 (20) | 74.06 (24) | 0.45 (22) | 3.88 (20) | 0.74 (24) | 26.53 | 7.95 | 34.48 |
| CP-034 | 19.68 (5) | 0 (53) | 1.14 (53) | 2.584 (14) | 99.42 (85) | 1 (53) | 19.68 (84) | 0 (53) | 0 (53) | 0 (53) | 9.84 (6) | 0.11 (58) | 0 (53) | 100 (53) | 0 (53) | 0 (53) | 1 (53) | 51.18 | 20.16 | 71.34 |
| CP-009 | 14.77 (17) | 1.976 (24) | 0.988 (37) | 2.362 (15) | 64.62 (73) | 0.86 (39) | 12.79 (72) | 0.29 (14) | 2.09 (14) | 5.40 (14) | 8.37 (15) | 0.36 (33) | 1.402 (24) | 86.61 (39) | 0.18 (34) | 3.48 (23) | 0.87 (39) | 32.76 | 18.24 | 51.00 |
| CP-001 | 9.36 (39) | 0 (53) | 1.14 (53) | 2.318 (16) | 47.28 (58) | 1 (53) | 9.36 (57) | 0 (53) | 0 (53) | 0 (53) | 4.68 (49) | 0.41 (29) | 0 (53) | 100 (53) | 0 (53) | 0 (53) | 1 (53) | 52.53 | 4.02 | 56.55 |
| CP-017 | 11.34 (26) | 0 (53) | 1.14 (53) | 2.151 (17) | 57.27 (68) | 1 (53) | 11.33 (67) | 0 (53) | 0 (53) | 0 (53) | 5.66 (34) | 0.31 (42) | 0 (53) | 100 (53) | 0 (53) | 0 (53) | 1 (53) | 52.06 | 9.76 | 61.82 |
| CP-019 | 2.74 (84) | 0.462 (52) | 0.949 (34) | 2.092 (18) | 11.52 (9) | 0.831 (37) | 2.28 (9) | 0.012 (52) | 0.09 (52) | 1.12 (52) | 1.60 (84) | 0.60 (16) | 0.328 (52) | 83.14 (37) | 0.055 (45) | 0.79 (52) | 0.83 (37) | 44.82 | 19.95 | 64.77 |
| CP-033 | 8.93 (43) | 0 (53) | 1.14 (53) | 2.086 (19) | 45.10 (55) | 1 (53) | 8.92 (54) | 0 (53) | 0 (53) | 0 (53) | 4.46 (52) | 0.74 (7) | 0 (53) | 100 (53) | 0 (53) | 0 (53) | 1 (53) | 52.59 | 2.58 | 55.16 |
| CP-045 | 8.08 (53) | 0 (53) | 1.14 (53) | 2.048 (20) | 40.82 (49) | 1 (53) | 8.08 (48) | 0 (53) | 0 (53) | 0 (53) | 4.04 (62) | 0.32 (40) | 0 (53) | 100 (53) | 0 (53) | 0 (53) | 1 (53) | 53.06 | 2.66 | 55.72 |
| CP-060 | 8.32 (48) | 1.92 (26) | 0.878 (26) | 1.999 (21) | 32.33 (39) | 0.769 (29) | 6.40 (39) | 0.163 (31) | 1.14 (31) | 3.99 (31) | 5.12 (40) | 0.92 (3) | 1.362 (26) | 76.92 (29) | 0.31 (30) | 3.12 (27) | 0.77 (29) | 31.71 | 6.13 | 37.84 |
| CP-014 | 14.34 (19) | 3.36 (10) | 0.874 (24) | 1.985 (22) | 55.45 (66) | 0.765 (27) | 10.97 (65) | 0.49 (9) | 3.45 (8) | 6.94 (9) | 8.84 (14) | 0.54 (18) | 2.381 (9) | 76.56 (27) | 0.55 (18) | 5.44 (8) | 0.77 (27) | 23.29 | 17.81 | 41.11 |
| CP-041 | 5.59 (74) | 1.088 (40) | 0.92 (30) | 1.975 (23) | 22.75 (23) | 0.805 (33) | 4.50 (23) | 0.062 (44) | 0.42 (44) | 2.46 (44) | 3.34 (67) | 0.68 (11) | 0.772 (40) | 80.54 (33) | 0.15 (36) | 1.82 (41) | 0.81 (33) | 39.47 | 13.39 | 52.86 |
| CP-057 | 11.20 (27) | 3.337 (11) | 0.801 (16) | 1.821 (24) | 39.72 (47) | 0.702 (18) | 7.86 (46) | 0.38 (12) | 2.68 (11) | 6.11 (12) | 7.26 (23) | 0.43 (24) | 2.368 (11) | 70.20 (18) | 0.70 (12) | 5.14 (10) | 0.70 (18) | 19.41 | 11.36 | 30.77 |
| CP-007 | 8.28 (50) | 1.958 (25) | 0.871 (23) | 1.718 (25) | 31.94 (38) | 0.763 (26) | 6.33 (38) | 0.165 (30) | 1.16 (30) | 4.02 (30) | 5.11 (41) | 0.25 (47) | 1.389 (25) | 76.34 (26) | 0.32 (29) | 3.16 (26) | 0.76 (26) | 30.29 | 7.26 | 37.56 |
| CP-030 | 6.88 (57) | 1.328 (37) | 0.921 (31) | 1.661 (26) | 28.04 (28) | 0.807 (34) | 5.55 (28) | 0.093 (36) | 0.65 (36) | 3.02 (36) | 4.11 (59) | 0.13 (54) | 0.942 (37) | 80.69 (34) | 0.17 (35) | 2.22 (37) | 0.81 (34) | 36.88 | 8.42 | 45.30 |

*(Continued)*

**Table 4.** (Continued)

| Genotype | Ypi | Ysi$_2$ | SSI | RDI | SSPI | SDI | TOL | STI | REI | GMP | MP | YSI | YI | PYR | DTI | HM | DII | Rank Mean | SD of Rank | Rank Sum |
|---|---|---|---|---|---|---|---|---|---|---|---|---|---|---|---|---|---|---|---|---|
| CP-020 | 15.44 (12) | 3.504 (6) | 0.882 (27) | 1.646 (27) | 60.30 (70) | 0.773 (30) | 11.93 (69) | 0.54 (7) | 3.87 (6) | 7.35 (7) | 9.47 (8) | 0.05 (64) | 2.487 (6) | 77.30 (30) | 0.56 (17) | 5.71 (5) | 0.77 (30) | 23.06 | 20.51 | 43.57 |
| CP-013 | 6.04 (70) | 1.72 (32) | 0.816 (19) | 1.641 (28) | 21.82 (19) | 0.715 (21) | 4.33 (19) | 0.106 (35) | 0.74 (35) | 3.22 (35) | 3.88 (65) | 0.46 (21) | 1.220 (32) | 71.52 (21) | 0.34 (28) | 2.67 (32) | 0.72 (21) | 30.94 | 15.14 | 46.08 |
| CP-012 | 14.56 (18) | 0(53) | 1.14 (53) | 1.621 (29) | 73.56 (78) | 1(53) | 14.56 (77) | 0(53) | 0(53) | 0(53) | 7.28 (22) | 0.36 (34) | 0(53) | 100 (53) | 0(53) | 0(53) | 1 (53) | 52.06 | 14.62 | 66.68 |
| CP-046 | 7.80 (55) | 2.872 (17) | 0.721 (13) | 1.593 (30) | 24.92 (26) | 0.632 (14) | 4.93 (26) | 0.228 (24) | 1.60 (24) | 4.73 (24) | 5.33 (37) | 0.95 (1) | 2.037 (17) | 63.20 (14) | 0.74 (11) | 4.19 (16) | 0.63 (14) | 21.18 | 11.05 | 32.23 |
| CP-003 | 9.58 (37) | 0.64 (49) | 1.065 (48) | 1.457 (31) | 45.16 (56) | 0.933 (48) | 8.93 (55) | 0.063 (43) | 0.43 (43) | 2.47 (43) | 5.10 (42) | 0.04 (66) | 0.454 (49) | 93.31 (48) | 0.030 (49) | 1.19 (48) | 0.93 (48) | 46.65 | 4.89 | 51.53 |
| CP-005 | 4.57 (77) | 0(53) | 1.14 (53) | 1.406 (32) | 23.07 (24) | 1(53) | 4.56 (24) | 0(53) | 0(53) | 0(53) | 2.28 (79) | 0.30 (44) | 0(53) | 100 (53) | 0(53) | 0(53) | 1 (53) | 52.53 | 13.53 | 66.06 |
| CP-004 | 25.04 (2) | 0.712 (45) | 1.109 (52) | 1.366 (34) | 122.91 (87) | 0.971 (52) | 24.32 (86) | 0.182 (29) | 1.27 (29) | 4.22 (29) | 12.87 (3) | 0.12 (56) | 0.505 (45) | 97.15 (52) | 0.014 (52) | 1.38 (44) | 0.97 (52 | 44.94 | 22.71 | 67.65 |
| CP-054 | 11.52 (25) | 3.36 (9) | 0.808 (18) | 1.355 (32) | 41.22 (50) | 0.708 (20) | 8.16 (49) | 0.39 (11) | 2.77 (10) | 6.22 (11) | 7.45 (20) | 0.02 (67) | 2.384 (10) | 70.83 (20) | 0.69 (13) | 5.20 (9) | 0.71 (20) | 19.82 | 12.41 | 32.23 |
| CP-021 | 10.35 (31) | 5.936 (3) | 0.487 (5) | 1.293 (35) | 22.31 (22) | 0.426 (5) | 4.41 (22) | 0.62 (5) | 4.40 (4) | 7.83 (5) | 8.14 (17) | 0.70 (9) | 4.215 (3) | 42.65 (5) | 2.41 (3) | 7.54 (4) | 0.43 (5) | 8.82 | 8.52 | 17.35 |
| CP-027 | 16.22 (10) | 3.368 (8) | 0.904 (28) | 1.186 (36) | 64.95 (74) | 0.792 (31) | 12.85 (73) | 0.55 (6) | 3.91 (5) | 7.39 (6) | 9.79 (7) | 0.08 (60) | 2.393 (8) | 79.24 (31) | 0.49 (20) | 5.57 (7) | 0.79 (31) | 24.06 | 21.70 | 45.76 |
| MTA-22 | 8.48 (46) | 2.8 (19) | 0.764 (14) | 2.318 (16) | 28.69 (30) | 0.669 (16) | 5.68 (30) | 0.242 (22) | 1.70 (22) | 4.87 (22) | 5.64 (35) | 0.92 (2) | 1.986 (19) | 66.98 (16) | 0.65 (15) | 4.20 (15) | 0.67 (16) | 21.71 | 8.77 | 30.48 |
| CP-036 | 16.62 (8) | 0(53) | 1.14 (53) | 0(53) | 83.98 (84) | 1(53) | 16.62 (83) | 0(53) | 0(53) | 0(53) | 8.31 (16) | 0 (69) | 0(53) | 100 (53) | 0(53) | 0(53) | 1 (53) | 51.82 | 18.19 | 70.01 |
| CP-026 | 16.72 (7) | 1.52 (35) | 1.037 (45) | 0.638 (45) | 76.79 (82) | 0.909 (45) | 15.20 (81) | 0.259 (18) | 1.82 (18) | 5.04 (18) | 9.12 (11) | 0.12 (57) | 1.078 (35) | 90.90 (45) | 0.098 (41) | 2.78 (31) | 0.91 (45) | 38.12 | 21.11 | 59.23 |
| CP-025 | 2.21 (86) | 0.656 (47) | 0.802 (17) | 2.086 (19) | 7.85 (3) | 0.702 (19) | 1.56 (3) | 0.014 (51) | 0.10 (51) | 1.20 (51) | 1.43 (86) | 0.32 (41) | 0.465 (47) | 70.28 (19) | 0.138 (37) | 1.01 (50) | 0.70 (19) | 31.41 | 24.67 | 56.08 |
| CP-069 | 8.38 (47) | 0.992 (43) | 1.006 (43) | 0.831 (43) | 37.34 (45) | 0.882 (43) | 7.39 (44) | 0.085 (38) | 0.59 (38) | 2.88 (38) | 4.69 (48) | 0.29 (45) | 0.703 (43) | 88.16 (43) | 0.083 (43) | 1.77 (42) | 0.88 (43) | 46.88 | 6.92 | 53.80 |
| CP-029 | 8.24 (51) | 0(53) | 1.14 (53) | 0(53) | 41.63 (51) | 1(53) | 8.24 (50) | 0(53) | 0(53) | 0(53) | 4.12 (58) | 0.37 (32) | 0(53) | 100 (53) | 0(53) | 0(53) | 1 (53) | 52.94 | 1.52 | 54.46 |
| CP-049 | 5.91 (71) | 4.36 (5) | 0.299 (3) | 5.179 (3) | 7.84 (2) | 0.262 (3) | 1.55 (2) | 0.263 (17) | 1.84 (17) | 5.07 (17) | 5.13 (39) | 0.06 (62) | 3.092 (5) | 26.25 (3) | 2.28 (4) | 5.01 (11) | 0.26 (3) | 12.06 | 18.02 | 30.07 |
| CP-008 | 6.40 (62) | 1.8 (29) | 0.820 (21) | 1.975 (23) | 23.24 (25) | 0.718 (23) | 4.60 (25) | 0.117 (34) | 0.82 (34) | 3.39 (34) | 4.10 (60) | 0 (69) | 1.277 (29) | 71.87 (23) | 0.35 (27) | 2.80 (30) | 0.72 (23) | 30.88 | 12.11 | 42.99 |
| CP-048 | 6.51 (61) | 3.016 (15) | 0.612 (7) | 3.252 (7) | 17.66 (14) | 0.536 (7) | 3.49 (14) | 0.200 (27) | 1.40 (27) | 4.43 (27) | 4.76 (47) | 0.62 (14) | 2.140 (15) | 53.68 (7) | 0.99 (7) | 4.12 (18) | 0.54 (7) | 19.41 | 14.94 | 34.35 |
| CP-068 | 6.72 (59) | 2.968 (16) | 0.977 (36) | 3.102 (9) | 18.95 (15) | 0.558 (9) | 3.75 (15) | 0.203 (25) | 1.43 (25) | 4.46 (25) | 4.84 (45) | 0.83 (5) | 2.106 (16) | 55.83 (9) | 0.93 (8) | 4.11 (19) | 0.56 (9) | 20.53 | 14.37 | 34.90 |
| CP-016 | 18.72 (6) | 14.77 (1) | 0.24 (2) | 5.543 (2) | 19.92 (17) | 0.211 (2) | 3.94 (17) | 2.82 (1) | 19.83 (1) | 16.63 (1) | 16.74 (2) | 0.80 (6) | 10.48 (1) | 21.06 (2) | 8.27 (1) | 16.51 (1) | 0.21 (2) | 3.00 | 4.00 | 7.00 |
| CP-031 | 3.26 (82) | 0(53) | 1.14 (53) | 0(53) | 16.49 (13) | 1(53) | 3.26 (13) | 0(53) | 0(53) | 0(53) | 1.63 (83) | 0.71 (8) | 0(53) | 100 (53) | 0(53) | 0(53) | 1 (53) | 51.76 | 17.53 | 69.29 |
| CP-028 | 8.30 (49) | 3.24 (14) | 0.696 (11) | 2.740 (12) | 25.58 (27) | 0.609 (12) | 5.06 (27) | 0.27 (16) | 1.92 (16) | 5.18 (16) | 5.77 (33) | 0.43 (23) | 2.299 (14) | 60.98 (12) | 0.89 (9) | 4.66 (14) | 0.61 (12) | 18.00 | 10.36 | 28.36 |
| CP-024 | 8.24 (52) | 0.648 (48) | 1.052 (47) | 0.552 (47) | 38.35 (46) | 0.921 (47) | 7.59 (45) | 0.054 (46) | 0.38 (46) | 2.31 (46) | 4.44 (54) | 0.42 (26) | 0.459 (48) | 92.13 (47) | 0.036 (47) | 1.20 (47) | 0.92 (47) | 47.53 | 2.18 | 49.71 |
| C | 4.00 (78) | 0(53) | 1.14 (53) | 0(53) | 20.20 (18) | 1(53) | 4.00 (18) | 0(53) | 0(53) | 0(53) | 2.00 (81) | 0.69 (10) | 0(53) | 100 (53) | 0(53) | 0(53) | 1 (53) | 52.00 | 15.50 | 67.50 |
| D | 6.07 (69) | 0(53) | 1.14 (53) | 0(53) | 30.67 (33) | 1(53) | 6.07 (33) | 0(53) | 0(53) | 0(53) | 3.03 (74) | 0.86 (4) | 0(53) | 100 (53) | 0(53) | 0(53) | 1 (53) | 52.82 | 9.67 | 62.49 |

(*Continued*)

**Table 4.** (Continued)

| Genotype | Ypi | Ysi$_2$ | SSI | RDI | SSPI | SDI | TOL | STI | REI | GMP | MP | YSI | YI | PYR | DTI | HM | DII | Rank Mean | SD of Rank | Rank Sum |
|---|---|---|---|---|---|---|---|---|---|---|---|---|---|---|---|---|---|---|---|---|
| NDT-003 | 3.81 (79) | 0(53) | 1.14 (53) | 0(53) | 19.23 (16) | 1(53) | 3.80 (16) | 0(53) | 0(53) | 0(53) | 1.90 (82) | 0.42 (27) | 0(53) | 100 (53) | 0(53) | 0(53) | 1 (53) | 51.88 | 16.27 | 68.15 |
| F | 6.22 (68) | 0(53) | 1.14 (53) | 0(53) | 31.44 (34) | 1(53) | 6.22 (34) | 0(53) | 0(53) | 0(53) | 3.11 (73) | 0 (69) | 0(53) | 100 (53) | 0(53) | 0(53) | 1 (53) | 52.82 | 9.17 | 62.00 |
| B | 6.24 (67) | 0(53) | 1.14 (53) | 0(53) | 31.53 (36) | 1(53) | 6.25 (36) | 0(53) | 0(53) | 0(53) | 3.12 (72) | 0 (69) | 0(53) | 100 (53) | 0(53) | 0(53) | 1 (53) | 52.94 | 8.42 | 61.36 |
| NDT-010 | 14.98 (14) | 3.5 (7) | 0.874 (25) | 1.641 (28) | 57.97 (69) | 0.766 (28) | 11.47 (68) | 0.53 (8) | 3.75 (7) | 7.23 (8) | 9.23 (9) | 0.33 (38) | 2.483 (7) | 76.62 (28) | 0.58 (16) | 5.67 (6) | 0.77 (28) | 22.71 | 19.62 | 42.33 |
| NDT-016 | 3.62 (80) | 1.616 (34) | 0.631 (8) | 3.138 (8) | 10.10 (7) | 0.553 (8) | 2.00 (7) | 0.059 (45) | 0.41 (45) | 2.41 (45) | 2.61 (78) | 0.41 (28) | 1.146 (34) | 55.30 (8) | 0.51 (19) | 2.23 (36) | 0,55 (8) | 28.12 | 24.46 | 52.57 |
| NDT-045 | 6.80 (58) | 1.144 (39) | 0.949 (34) | 1.181 (38) | 28.57 (29) | 0.831 (38) | 5.65 (29) | 0.079 (40) | 0.55 (40) | 2.78 (40) | 3.97 (63) | 0.63 (13) | 0.811 (39) | 83.17 (38) | 0.136 (38) | 1.95 (39) | 0.83 (38) | 39.94 | 8.47 | 48.41 |
| NDT-004 | 10.90 (29) | 1.392 (36) | 0.995 (42) | 0.897 (42) | 48.01 (60) | 0.872 (42) | 9.50 (59) | 0.154 (32) | 1.08 (32) | 3.89 (32) | 6.14 (30) | 0.13 (55) | 0.987 (36) | 87.22 (42) | 0.126 (39) | 2.46 (34) | 0.87 (42) | 39.53 | 9.02 | 48.55 |
| NDT-002 | 14.88 (16) | 0(53) | 1.14 (53) | 0(53) | 75.17 (80) | 1(53) | 14.88 (79) | 0(53) | 0(53) | 0(53) | 7.44 (21) | 0 (69) | 0(53) | 100 (53) | 0(53) | 0(53) | 1 (53) | 52.12 | 15.49 | 67.61 |
| E | 14.98 (15) | 0(53) | 1.14 (53) | 0(53) | 75.66 (81) | 1(53) | 14.97 (80) | 0(53) | 0(53) | 0(53) | 7.48 (19) | 0.06 (63) | 0(53) | 100 (53) | 0(53) | 0(53) | 1 (53) | 52.06 | 16.11 | 68.17 |
| C-9 | 42.63 (1) | 1.776 (30) | 1.094 (51) | 0.292 (51) | 206.41 (88) | 0.958 (51) | 40.85 (87) | 0.77 (4) | 5.42 (3) | 8.70 (4) | 22.20 (1) | 0.20 (48) | 1.260 (30) | 95.83 (51) | 0.052 (46) | 3.40 (25) | 0.96 (51) | 36.12 | 28.26 | 64.38 |
| C-7 | 23.38 (3) | 1.064 (41) | 1.089 (50) | 0.319 (50) | 112.72 (86) | 0.954 (50) | 22.31 (85) | 0.253 (19) | 1.78 (19) | 4.98 (19) | 12.22 (5) | 0.20 (49) | 0.755 (41) | 95.44 (50) | 0.034 (48) | 2.03 (38) | 0.95 (50) | 42.06 | 23.46 | 65.52 |
| NDT-029 | 3.20 (83) | 1.304 (38) | 0.676 (10) | 2.862 (11) | 9.57 (5) | 0.592 (11) | 1.89 (5) | 0.042 (47) | 0.29 (47) | 2.04 (47) | 2.25 (80) | 0.14 (53) | 0.925 (38) | 59.25 (11) | 0.37 (25) | 1.85 (40) | 0.59 (11) | 30.59 | 24.85 | 55.44 |
| C-14 | 10.18 (33) | 0(53) | 1.14 (53) | 0(53) | 51.45 (64) | 1(53) | 10.18 (63) | 0(53) | 0(53) | 0(53) | 5.09 (43) | 0.16 (51) | 0(53) | 100 (53) | 0(53) | 0(53) | 1 (53) | 52.53 | 6.79 | 59.32 |
| C-10 | 16.36 (9) | 1.704 (33) | 1.022 (44) | 0.731 (40) | 74.04 (79) | 0.895 (44) | 14.65 (78) | 0.28 (15) | 1.99 (15) | 5.27 (15) | 9.03 (12) | 0.42 (25) | 1.209 (33) | 89.58 (44) | 0.125 (40) | 3.08 (28) | 0.90 (44) | 37.76 | 20.62 | 58.38 |
| C-18 | 13.52 (20) | 0(53) | 1.14 (53) | 0(53) | 68.30 (76) | 1(53) | 13.52 (75) | 0(53) | 0(53) | 0(53) | 6.76 (25) | 0.31 (43) | 0(53) | 100 (53) | 0(53) | 0(53) | 1 (53) | 55.06 | 12.65 | 67.71 |
| C-19 | 9.60 (36) | 0(53) | 1.14 (53) | 0(53) | 48.50 (61) | 1(53) | 9.60 (60) | 0(53) | 0(53) | 0(53) | 4.80 (46) | 0.07 (61) | 0(53) | 100 (53) | 0(53) | 0(53) | 1 (53) | 52.53 | 5.37 | 57.90 |
| NDT-015 | 13.28 (21) | 0(53) | 1.14 (53) | 0(53) | 67.09 (75) | 1(53) | 13.28 (74) | 0(53) | 0(53) | 0(53) | 6.64 (26) | 0 (69) | 0(53) | 100 (53) | 0(53) | 0(53) | 1 (53) | 52.12 | 13.01 | 65.13 |
| NDP-017 | 9.36 (40) | 0(53) | 1.14 (53) | 0(53) | 47.29 (59) | 1(53) | 9.37 (58) | 0(53) | 0(53) | 0(53) | 4.67 (50) | 0 (69) | 0(53) | 100 (53) | 0(53) | 0(53) | 1 (53) | 52.76 | 3.95 | 56.71 |
| NDT-005 | 9.92 (34) | 0(53) | 1.14 (53) | 0(53) | 50.11 (62) | 1(53) | 9.92 (61) | 0(53) | 0(53) | 0(53) | 4.96 (44) | 0 (69) | 0(53) | 100 (53) | 0(53) | 0(53) | 1 (53) | 52.41 | 6.11 | 58.53 |
| NDT-006 | 5.61 (73) | 3.288 (12) | 0.472 (4) | 4.117 (4) | 11.72 (10) | 0.413 (4) | 2.32 (10) | 0.188 (28) | 1.32 (28) | 4.29 (28) | 4.45 (53) | 0.34 (37) | 2.333 (12) | 41.36 (4) | 1.36 (5) | 4.14 (17) | 0.41 (4) | 17.65 | 19.52 | 37.16 |
| NDT-025 | 8.80 (44) | 0(53) | 1.14 (53) | 0(53) | 44.45 (54) | 1(53) | 8.80 (53) | 0(53) | 0(53) | 0(53) | 4.40 (55) | 0.57 (17) | 0(53) | 100 (53) | 0(53) | 0(53) | 1 (53) | 52.71 | 2.31 | 55.02 |
| NDT-001 | 2.56 (85) | 0(53) | 1.14 (53) | 0(53) | 12.93 (11) | 1(53) | 2.56 (11) | 0(53) | 0(53) | 0(53) | 1.28 (87) | 0 (69) | 0(53) | 100 (53) | 0(53) | 0(53) | 1 (53) | 51.94 | 18.86 | 70.80 |
| A | 7.50 (56) | 1(42) | 0.989 (39) | 0.935 (40) | 32.85 (41) | 0.866 (40) | 6.50 (41) | 0.076 (41) | 0.53 (41) | 2.73 (41) | 4.25 (56) | 0.51 (19) | 0.709 (42) | 86.67 (40) | 0.094 (42) | 1.76 (43) | 0.87 (40) | 42.65 | 5.12 | 47.77 |
| NDT-009 | 12.21 (22) | 0(53) | 1.14 (53) | 0(53) | 61.67 (71) | 1(53) | 12.20 (70) | 0(53) | 0(53) | 0(53) | 6.10 (31) | 0 (69) | 0(53) | 100 (53) | 0(53) | 0(53) | 1 (53) | 53.71 | 13.52 | 67.23 |
| NDT-027 | 4.96 (75) | 0.64 (50) | 0.994 (41) | 0.906 (41) | 21.83 (20) | 0.871 (41) | 4.32 (20) | 0.032 (49) | 0.22 (49) | 1.78 (49) | 2.80 (76) | 0.44 (22) | 0.454 (50) | 87.09 (41) | 0.058 (44) | 1.13 (49) | 0.87 (41) | 45.71 | 14.39 | 60.10 |
| NDT-011 | 2.08 (87) | 0.85 (44) | 0.673 (9) | 2.879 (10) | 6.20 (1) | 0.59 (10) | 1.22 (1) | 0.018 (50) | 0.12 (50) | 1.33 (50) | 1.46 (85) | 0 (69) | 0.605 (44) | 59.00 (10) | 0.24 (32) | 1.21 (46) | 0.59 (10) | 32.29 | 27.57 | 59.86 |

(*Continued*)

**Table 4.** (Continued)

| Genotype | Ypi | Ysi$_2$ | SSI | RDI | SSPI | SDI | TOL | STI | REI | GMP | MP | YSI | YI | PYR | DTI | HM | DII | Rank Mean | SD of Rank | Rank Sum |
|---|---|---|---|---|---|---|---|---|---|---|---|---|---|---|---|---|---|---|---|---|
| NDT-008 | 11.98 (23) | 2.024 (23) | 0.948 (33) | 1.186 (36) | 50.32 (63) | 0.831 (36) | 9.96 (62) | 0.247 (20) | 1.73 (20) | 4.92 (20) | 7.00 (24) | 0.10 (59) | 1.436 (23) | 83.11 (36) | 0.23 (33) | 3.46 (24) | 0.83 (36) | 32.29 | 13.24 | 45.54 |
| C-8 | 6.59 (60) | 0(53) | 1.14 (53) | 0(53) | 33.30 (42) | 1(53) | 6.59 (41) | 0(53) | 0(53) | 0(53) | 3.29 (68) | 0.40 (30) | 0(53) | 100 (53) | 0(53) | 0(53) | 1 (53) | 53.00 | 5.68 | 58.68 |
| NDT-013 | 16.16 (11) | 0(53) | 1.14 (53) | 0(53) | 81.64 (83) | 1(53) | 16.16 (82) | 0(53) | 0(53) | 0(53) | 8.08 (18) | 0.01 (68) | 0(53) | 100 (53) | 0(53) | 0(53) | 1 (53) | 52.00 | 17.27 | 69.27 |
| NDT-022 | 3.54 (81) | 1.84 (28) | 0.547 (6) | 3.654 (6) | 8.56 (4) | 0.479 (6) | 1.69 (4) | 0.066 (42) | 0.46 (42) | 2.55 (42) | 2.68 (77) | 0 (69) | 1.305 (28) | 47.96 (6) | 0.67 (14) | 2.42 (35) | 0.48 (6) | 25.47 | 24.99 | 50.46 |
| NDT-018 | 4.80 (76) | 1.768 (31) | 0.721 (12) | 2.586 (13) | 15.31 (12) | 0.631 (13) | 3.03 (12) | 0.086 (37) | 0.60 (37) | 2.91 (37) | 3.28 (69) | 0 (69) | 1.254 (31) | 63.16 (13) | 0.46 (21) | 2.58 (33) | 0.63 (13) | 27.82 | 19.70 | 47.53 |
| NDT-026 | 9.10 (41) | 0(53) | 1.14 (53) | 0(53) | 45.99 (57) | 1(53) | 9.10 (56) | 0(53) | 0(53) | 0(53) | 4.55 (51) | 0 (69) | 0(53) | 100 (53) | 0(53) | 0(53) | 1 (53) | 52.65 | 3.33 | 55.98 |
| NDT-020 | 6.32 (63) | 0.504 (51) | 1.051 (46) | 0.560 (46) | 29.38 (32) | 0.920 (46) | 5.81 (32) | 0.033 (48) | 0.24 (48) | 1.79 (48) | 3.41 (66) | 0 (69) | 0.357 (51) | 92.02 (46) | 0.028 (51) | 0.93 (51) | 0.92 (46) | 48.65 | 7.47 | 56.12 |
| NDT-023 | 7.89 (54) | 0(53) | 1.14 (53) | 0(53) | 39.85 (48) | 1(53) | 7.88 (47) | 0(53) | 0(53) | 0(53) | 3.94 (64) | 0 (69) | 0(53) | 100 (53) | 0(53) | 0(53) | 1 (53) | 53.12 | 3.28 | 56.39 |
| NDT-012 | 5.79 (72) | 0(53) | 1.14 (53) | 0(53) | 29.26 (31) | 1(53) | 5.79 (31) | 0(53) | 0(53) | 0(53) | 2.89 (75) | 0 (69) | 0(53) | 100 (53) | 0(53) | 0(53) | 1 (53) | 52.82 | 10.64 | 63.47 |
| NDT-025 B | 6.32 (64) | 0(53) | 1.14 (53) | 0(53) | 31.93 (37) | 1(53) | 6.32 (37) | 0(53) | 0(53) | 0(53) | 3.16 (70) | 0 (69) | 0(53) | 100 (53) | 0(53) | 0(53) | 1 (53) | 52.76 | 7.59 | 60.35 |

Values in the parentheses refer to the ranks of the genotype for each index, YSi2 = Mean value of grain yield under drought stress condition, Ypi = mean value of yield under non-stress condition, SSI = Stress susceptibility index, RDI = relative drought index, SSPI = Stress susceptibility percentage index, SDI = Sensitivity drought index, TOL = stress tolerance, STI = Stress tolerance index, REI = The relative efficiency index, GMP = Geometric mean productivity, MP = Mean productivity, YSI = Yield stability index, YI = Yield index, PYR = percentage of yield reduction, DTI = Drought tolerance index, HM = Harmonic mean, DII = Drought Intensity Index

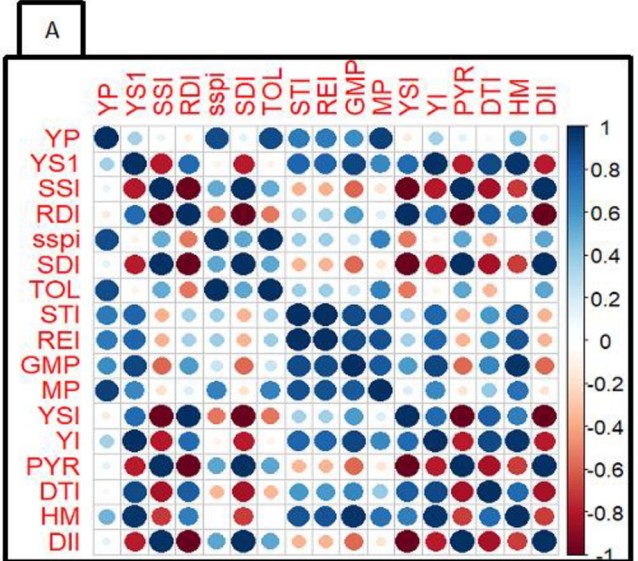
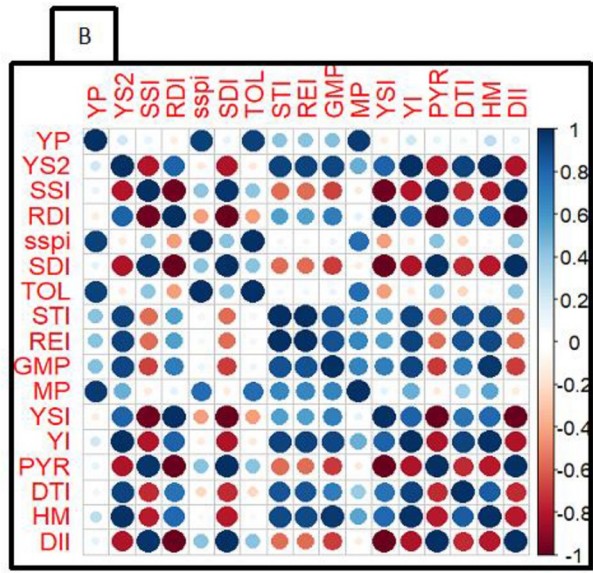

**Fig 1.** Pearson's correlation plot of yield under non-stress (YP), stress (YS) and fifteen drought tolerance indices under VDS (A) and FDS (B) in cowpea genotypes. Pairwise correlations between variables are colour-mapped according to the colour scale of the bar presented at the right-hand side.

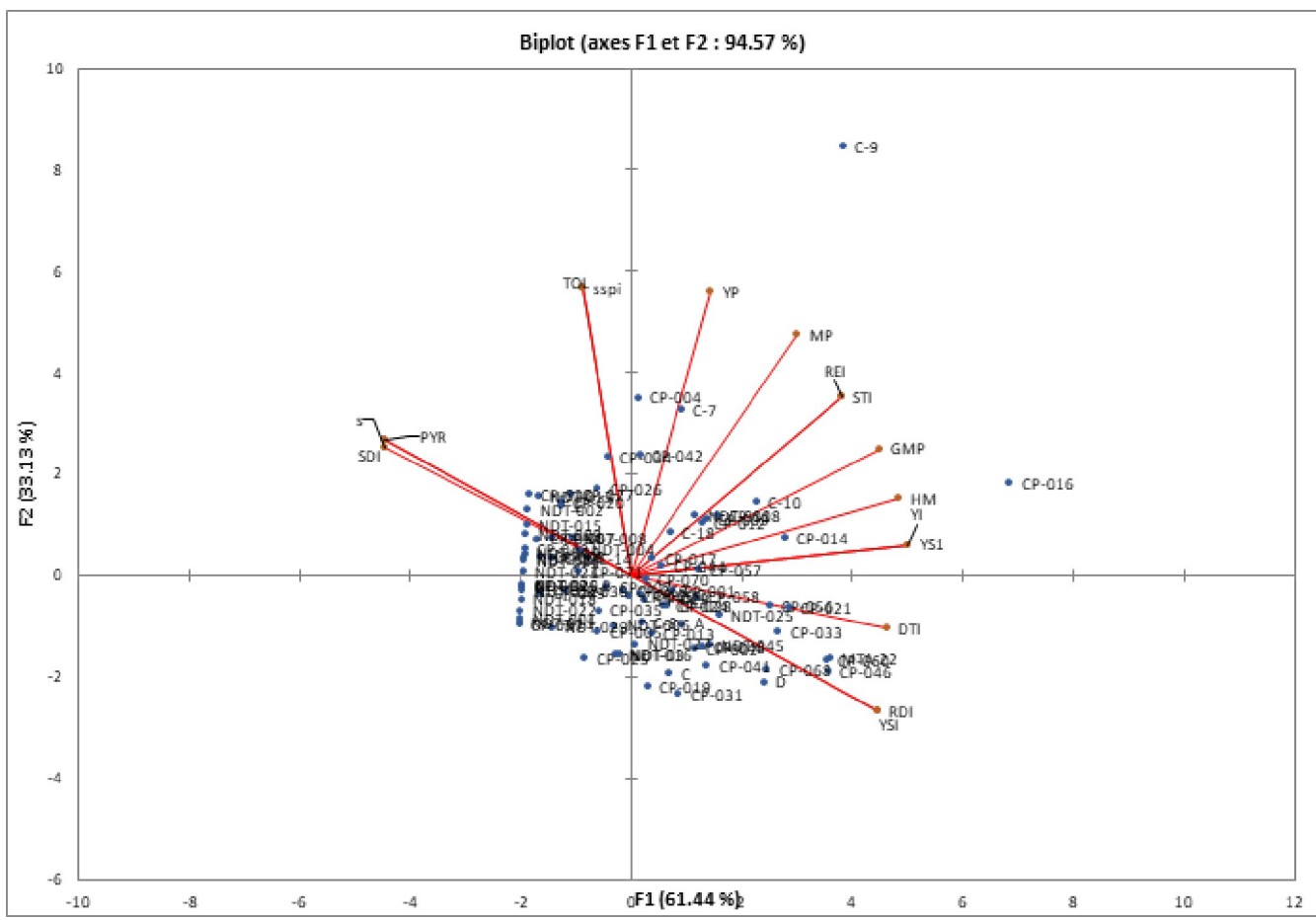

**Fig 2. Biplot for drought tolerance indices based on the first two principal components axes (PC1 and PC2) for eighty-eight cowpea genotypes under VDS.**

variation (Fig 2). The first two components under FDS corresponded to the essential variance in cowpea genotypes, accounting for 92.42% cumulatively (PC1 = 64.44% and PC2 = 27.98%, Fig 3). The indices STI, REI, GMP, MP, YI, HM, Ys, and Yp contributed positively and significantly to PC1 for both stresses (Data not shown). These indices were strongly associated with yield under stress and non-stress conditions and therefore, found as the most suitable indices to screen high-yielding and drought-tolerant genotypes under both stress and non-stress conditions. Consequently, the PC1 could be baptised component of tolerance to drought stress and associated genotypes from the biplot analysis are to be high yielding under non-stress and stress conditions. As a result, genotypes CP-016, C-9, MTA-22, CP-046, CP-060 for VDS phase and CP-016, CP-056, CP-021, CP-049, CP-040, NDT-006 for FDS stage with the higher and significant PC1 scores associated with irrelevant PC2 values were superior plants under both non-stress and moisture stress conditions (Figs 2 and 3). Meanwhile, under the two moisture stress regimes, Genotypes with larger PC2 scores associated with insignificant PC1 values performed poorly. These were CP-004, C7, CP-042, CP-034, CP-031 and CP-019 for VDS stage and C-9, C-7, CP-004, NDT-011, NDT-006, CP-034 and CP-031 for FDS stage (Figs 2 and 3). Considering yield in stressed conditions (Ys), the best index revealed by the biplot analysis was YI as their association was also strong and positive (Fig 1). This show that genotypes CP-016, C-9, CP-014, MTA-22 under VDS (Fig 2) and CP-016, CP056, CP-021, CP-042

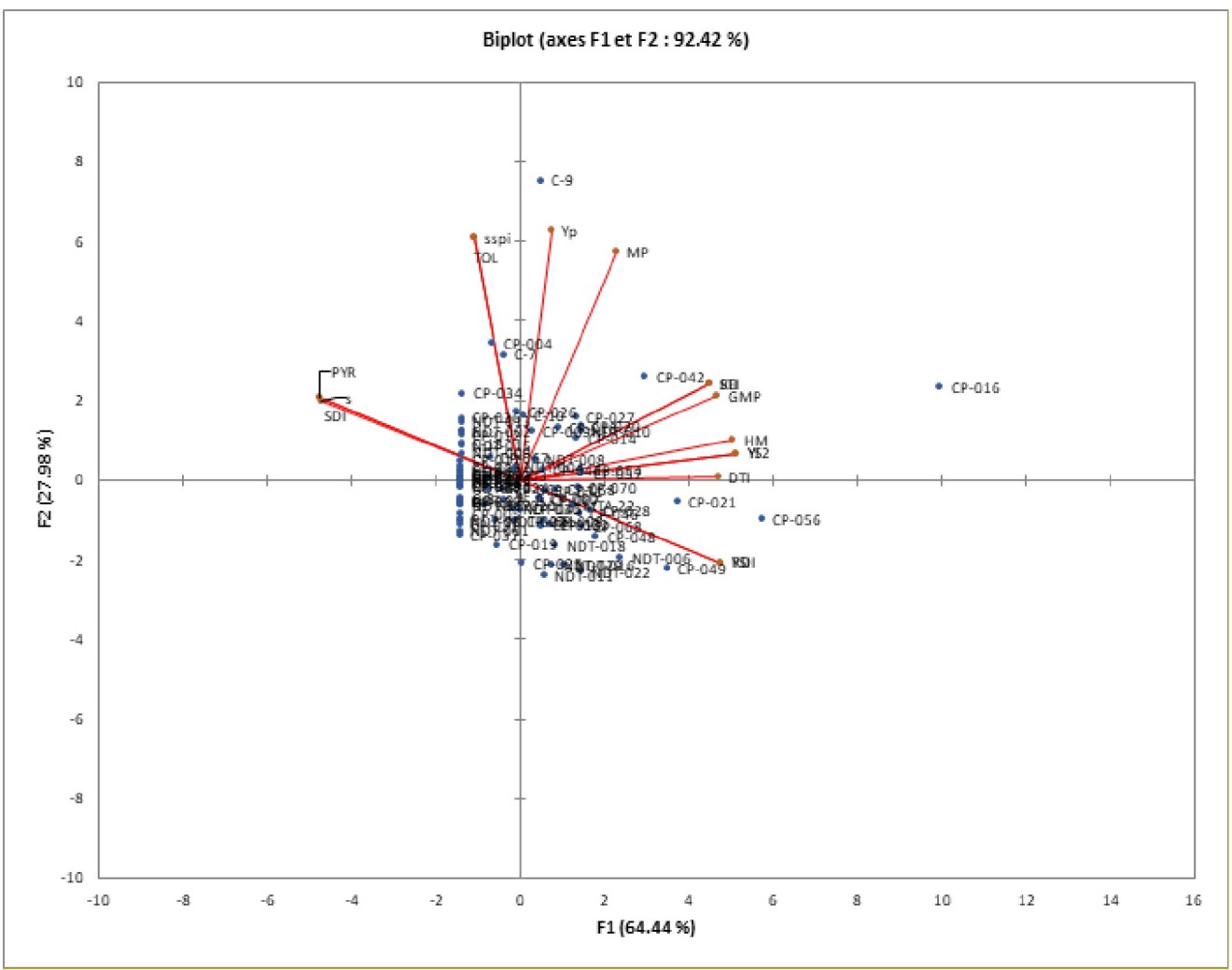

**Fig 3. Biplot for drought tolerance indices based on the first two principal components axes (PC1 and PC2) for eighty-eight cowpea genotypes under FDS.**

under FDS (Fig 3) are best suited in stress environments. In contrast, large obtuse angles between the vectors of SSI, SDI, and PRY indices with Ys, indicated a strong negative association between those indices and Ys. The bi-plot displayed a series of genotypes with irrelevant Ys scores (like CP-037, NDT-013, CP-054, CP-003, CP-067, CP-020) as highly drought sensitive with low yield stability. Biplots analysis also revealed a strong negative association between SSI, DTI, and PYR on one hand with YSI and RDI values on the other hand. The highest YSI and RDI values were obtained by genotypes CP-046 MTA-22, CP-060, D, CP-068, CP-016 and CP-033 under VDS and CP-046, CP-016, CP-049, NDT-006 under FDS, showing high grain yield stability and important tolerance ranking.

## Cluster analysis

Cluster analysis based on fifteen drought tolerance indices and grain yield under VDS and FDS conditions produced each a dendrogram that divided the eighty-eight cowpea genotypes in three main clusters. Under VDS (Fig 4), cluster 1 was the largest, containing 51 genotypes

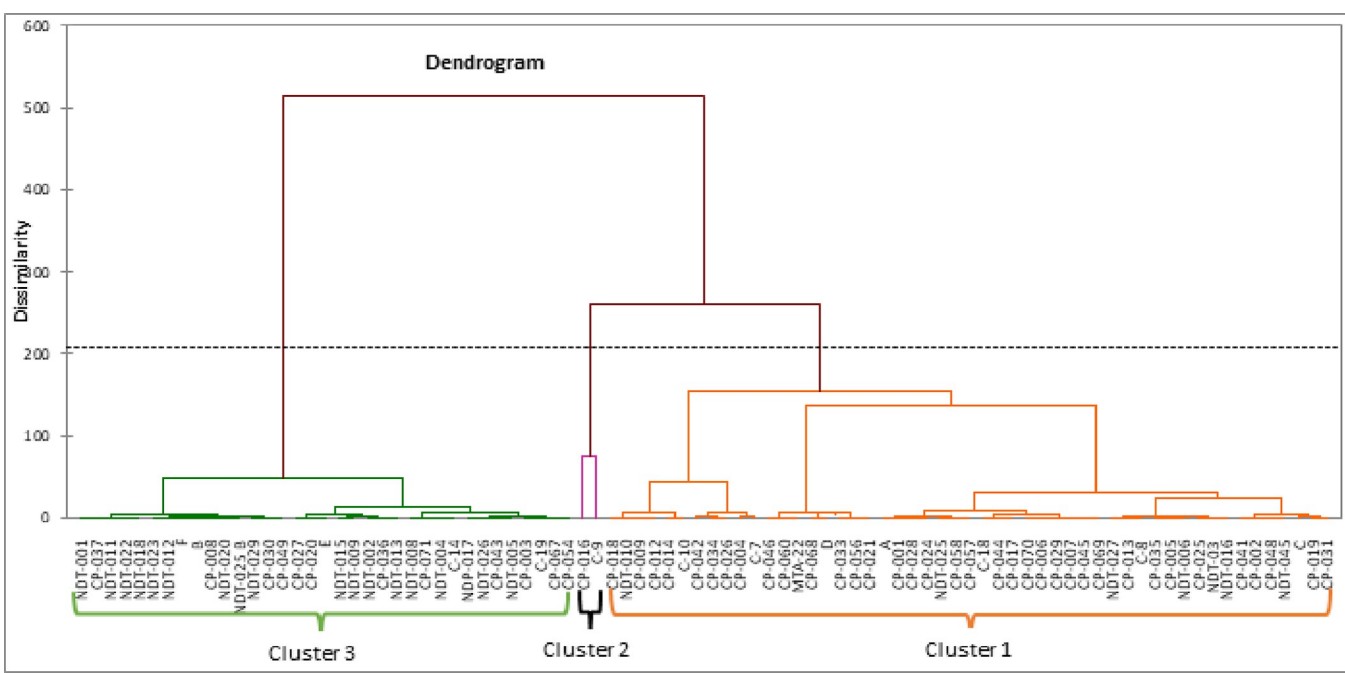

**Fig 4. Dendrogram grouping the eighty-eight cowpea genotypes using average linkage method based on fifteen drought tolerance indices under VDS.**

(57.95%) followed by cluster 3, which contained 35 genotypes (39.77%) and cluster 2 was the smallest group containing two genotypes (2.27%). While under FDS, cluster 3 was the largest, containing 51 genotypes (57.95%) followed by cluster 1, with 36 genotypes (41.00%) and cluster 2 consisted of a single genotype (1.13%) (Fig 5). Genotypes grouped in cluster 1 for both VDS and FDS were characterized in general by highest YSI, RDI and moderate Ys, and YI indices (Tables 5 and 6) and was considered a stable group under both stress and non-stress conditions. Genotypes grouped in this cluster were identified as tolerant or semi-tolerant to moisture stress. From this group, subgroups have been distinguished, among which genotypes with moderate to high grain yield with high stability under both moisture stress conditions like CP-046, MTA-22, CP-060, D for VDS and CP-056, CP-021, CP-049 under FDS. Another sub-group clustered genotypes with least stability performance like CP-034, CP-002, C-10, CP-014 for VDS and MTA-22, CP-026 under FDS. Under VDS, cluster 2 had two genotypes (CP-016, C-9) and CP016 was the only genotype for Cluster 2 under FDS. These genotypes, principally CP-016 had higher performance in both stressed and non-stressed environments than any other genotypes tested in this study. Moreover, these genotypes had high grain yield under both moisture regimes as well as reliable drought tolerance indices such as STI, REI, GMP, MP, YI, DTI and HM (Tables 5 and 6). Therefore, this group was considered as having the most desirable genotype for both growth conditions and tolerant to moisture stress. Genotypes grouped in cluster 3 were characterized by poor-yield performance under moisture stress conditions. From this clusters, two others subgroups were identified. The first sub-groups clustered for example NDT-013, CP-036, NDT-015 and NDT-009 for VDS, C-9, CP-004, C-7 and CP-034 for FDS. These genotypes were characterized by high to moderate performance under non-moisture stress conditions and had high values of SSI and important TOL and SSPI indices (Tables 5 and 6). They were susceptible to drought and suitable for non-stress conditions only. Under VDS, genotypes named CP-037, NDT-001, NDT-011, NDT-029 and CP-019, CP-025, CP-029, NDT-009 under FDS were clustered together in one subgroup and had the lowest

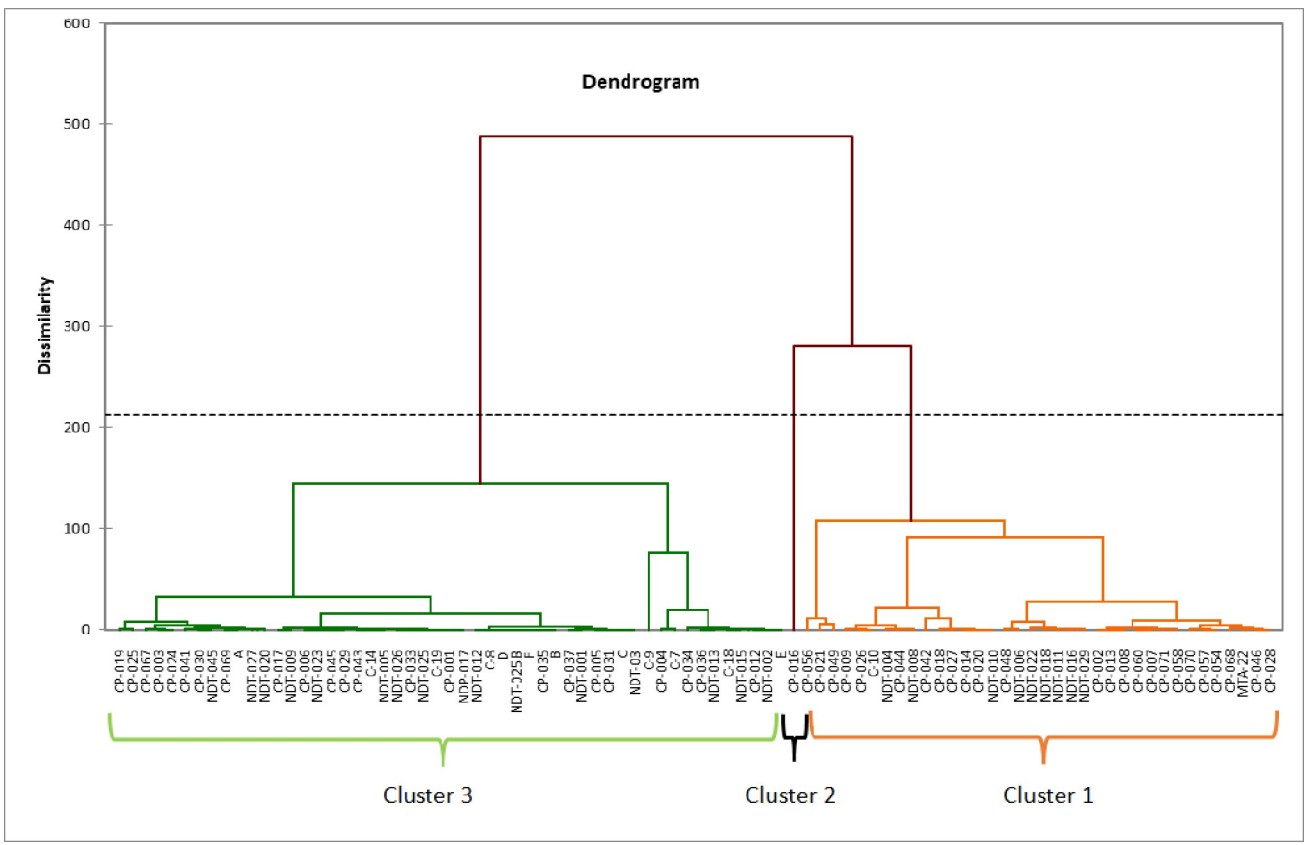

**Fig 5. Dendrogram grouping the eighty-eight cowpea genotypes using average linkage method based on fifteen drought tolerance indices under FDS.**

values for STI, HM, GMP, MP, Ys, and Yp (Tables 5 and 6). Hence, these genotypes performed poorly under both stress and non-stress conditions and were also identified as highly sensitive genotypes to drought with least stability performance.

## Discussion

According to the literature, moderate to high stress levels are useful and considered ideal for genotypic selection as insufficient stress could result in selection of non-resistant genotypes [53–55]. Cowpea genotypes in this study were evaluated under severe drought stress as shown a drought intensity index of 0.71 and 0.84 respectively for drought stress initiated at the

**Table 5. Mean values of yield under non-stress (YP), stress (YS1) and fifteen drought tolerance indices for the three clusters revealed by cluster analysis on eighty-eight cowpea genotypes for VDS.**

| Cluster | YP | YS1 | SSI | RDI | sspi | SDI | TOL | STI | REI | GMP | MP | YSI | YI | PYR | DTI | HM | DII |
|---|---|---|---|---|---|---|---|---|---|---|---|---|---|---|---|---|---|
| Cluster 1 | 9.61[b] | 3.94[b] | 0.71[b] | 1.73[a] | 28.67[c] | 0.54[b] | 5.67[b] | 0.41[b] | 1.54[b] | 5.93[b] | 6.78[b] | 0.46[a] | 1.48[b] | 53.51[b] | 0.77[b] | 5.28[b] | 0.54[b] |
| Cluster 2 | 30.68[a] | 11.72[a] | 0.65[b] | 1.86[a] | 95.75[a] | 0.50[b] | 18.95[a] | 3.27[a] | 12.13[a] | 17.85[a] | 21.20[a] | 0.50[a] | 4.40[a] | 49.96[b] | 2.58[a] | 15.36[a] | 0.50[b] |
| Cluster 3 | 9.12b | 0.35[c] | 1.26[a] | 0.13[b] | 44.34[b] | 0.96[a] | 8.78[b] | 0.04[c] | 0.15[c] | 1.21[c] | 4.73[b] | 0.04[b] | 0.13[c] | 96.47[a] | 0.01[c] | 0.63[c] | 0.96[a] |
| ANOVA Test | ** | ** | ** | ** | ** | ** | ** | ** | ** | ** | ** | ** | ** | ** | ** | ** | ** |

Values with different letters in the same column indicate a significant difference at 0.050 probability level.

**: Significant at p = 0.010

**Table 6. Mean values of yield under non-stress (YP), stress (YS2) and fifteen drought tolerance indices for the three clusters revealed by cluster analysis on eighty-eight cowpea genotypes for FDS.**

| Cluster | YP | YS2 | SSI | RDI | sspi | SDI | TOL | STI | REI | GMP | MP | YSI | YI | PYR | DTI | HM | DII |
|---|---|---|---|---|---|---|---|---|---|---|---|---|---|---|---|---|---|
| Cluster 1 | 9.75[b] | 2.80[b] | 0.79[b] | 2.30[b] | 35.09[b] | 0.67[b] | 6.95[ab] | 0.30[b] | 2.10[b] | 5.06[b] | 6.27[b] | 0.33[b] | 1.99[b] | 67.20[b] | 0.74[b] | 4.15[b] | 0.67[ab] |
| Cluster 2 | 18.72[a] | 14.78[a] | 0.24[c] | 5.54[a] | 19.93[c] | 0.21[c] | 3.94[b] | 2.82[a] | 19.83[a] | 16.63[a] | 16.75[a] | 0.79[a] | 10.49[a] | 21.07[c] | 8.28[a] | 16.52[a] | 0.21[b] |
| Cluster 3 | 9.83[b] | 0.26[c] | 1.10[a] | 0.25[c] | 48.33[a] | 0.96[a] | 9.57[a] | 0.04[b] | 0.26[c] | 0.89[b] | 5.05[b] | 0.04c | 0.19[c] | 96.47[a] | 0.02[b] | 0.47[c] | 0.96[a] |
| ANOVA Test | ** | ** | ** | ** | * | ** | * | ** | ** | ** | ** | ** | ** | ** | ** | ** | ** |

Values with different letters in the same column indicate a significant difference at 0.050 probability level.

*: Significant at p = 0.050.

**: Significant at p = 0.010

vegetative and flowering stages. These high indices revealed the significant impact of drought applied at both stress stages compared non-stress conditions. Both drought stresses were suitable for this study, revealing genotypic differences as seen by differential responses for yield in the studied genotypes. This is agreement with the study of Pejic et al. [26] on cowpea and Darkwa et al. [35] in common bean who also reported genotypic variation in their studied accessions under drought stress. This study revealed severe reduction (up to almost 85%) in yields from unstressed plants for stress during the vegetative and flowering stages. Ambachew et al. [29] and Beebe et al. [56] reported similar results when studying common bean with drought stress reducing seed yield to very low levels. Drought intensity index (DII) was significantly higher under FDS compared to VDS. The same trend was observed for SDI, SSPI and PYR with values under FDS significantly higher compared to values obtained under VDS. This reveals the sensitiveness of the flowering stage to drought compared to the vegetative stage and confirms previous reports on cowpea by Turk et al. [25] and wheat by Bahar and Yildirim [57] who showed that drought stress negatively and significantly affects flowering and grain filling with considerable yield reduction. Lisar et al. [58] reported 50% yield reduction compared to control plants for accessions under water stress during the setting of flowers. Oury et al. [59] also observed that water deficit conditions are responsible for large number of flower abortions with subsequent reduction of crop yield in maize. The severity of this stress (FDS) compared to VDS is because it directly affects parameters related to seed formation that include photosynthesis and amino acid metabolism [60, 61].

Understanding of the relationships among yield-based indices in connection to drought tolerance under drought-stress conditions provides to plant breeders desirable drought tolerant selection criteria. Yield under non-stress conditions (Yp) and yield under drought stress conditions (Ys) were positively and significantly correlated. This positive and significant relationship was also reported by Fernandez [42], Farshadfar and Elyasi [62] and Ferede et al. [63]. Significant correlation between grain yields under moisture deficit conditions (Ys) with some indices indicating their usefulness for selection of genotypes expressing uniform and high grain yield under both stressed and non-stressed conditions. Significant association were found between many couple of drought tolerance indices (Fig 1). We found in both VDS and FDS significant and positive correlation between mean grain yields under stress (Ys) with some indices on the one hand and significant and negative correlation recorded between Ys and some others stress indices on the other hand (Fig 1). These negative and significant associations are indicative that selection for some low indices value enable selection for high yield under drought stressed condition. These observations are similar to those found when studying maize under drought stress by Naghavi et al. [64] and wheat landrace exposed to moisture deficit conditions by Farshadfar and Elyasi [62], Moosavi et al. [45]. Susceptibility of cowpea

genotypes to drought stress based on a single criterion does not provide conclusive result as different indices may identify different genotypes as drought tolerant. For example: at the vegetative stage, the highest SSPI and TOL showed that C-9 is among the sensitive genotype while according to the STI, GM, MP, YI, REI and HM index, genotype C-9 was considered as one of the drought tolerant genotype. Therefore, the differentiation and separation of genotypes with different levels of drought tolerance was carried out taking into account all tolerance indices as also carried out Pejic et al. [26] in cowpea, Naghavi et al. [64] and Shojaei et al. [38] in maize, Darkwa et al. [35] in common bean and Yahaya et al. [41] in sorghum when differentiating susceptible from tolerant genotypes using yield-based drought tolerant indices.

From STI, REI, GMP, MP, YI and HM indices, biplot diagram of the principal component analysis (PCA) and correlation data revealed positive and significant association of these indices with yield of genotypes under both stress and non-stress conditions. Mau et al [65] and Hussain et al. [66] in rice, Farshadfar and Elyasi [62] in wheat and Naghavi et al. [64] in maize reported the useful of these indices as tools in selecting drought tolerant and high yielding genotypes. Based on YSI index, some genotypes evaluated to a stressed environments were suited for the stress environment while in contrast, a series of genotypes with low and valueless YSI were highly drought sensitive with low yield stability. Under VDS and FDS, high grain yield stability with moderate to high tolerance was obtained through YSI and RDI indices. Bouslama and Schapaugh [48] also concluded that cultivars with high YSI are expected to have high to moderate yield under stress and non-stress conditions, yield stability being more important than high yield as also concluded Moosavi et al. [45]. Cluster analysis based on the studied yield-based drought tolerance indices and grain yield under VDS and FDS produced each a dendogram with three main groups likely in link with tolerant, semi-tolerant and sensitive genotypes. Genotypes characterized by highest YSI, RDI, Ys, and YI indices were considered a stable group under both conditions. The genotypes in this group will have high stability with moderate to high grain yield (Genotypes in cluster 2 and most of genotypes in cluster 1 for both VDS and FDS). They were identified as tolerant or semi-tolerant genotypes to moisture stress. Similarly to our studies, Gavuzzi et al, [50] suggested the yield index (YI) while and Bouslama and Schapaugh [48] suggested the yield stability index (YSI) to evaluate the stability of genotypes in the both stress and non-stress conditions and identified tolerant genotypes. Cluster 2 contains two genotypes (CP-016 and C-9) for VDS and one genotype (CP-016) for FDS. These genotypes had a higher performance in both stress and non-stress environments than any other genotypes tested. Moreover, these genotypes had high grain yield under both moisture regimes as well as reliable drought tolerance indices such as STI, GMP, MP, YI, DTI and HM. Therefore, these genotypes grouped in cluster 2 can be identified as best performing genotypes under both non-stressed and stressed conditions and considered as the most desirable cluster with tolerant genotypes to moisture stress. Farshadfar and Elyasi [62] also identifies similar group of plants performing well under non-stressed and stressed conditions when studying wheat under drought stress. From Poor-yield performance genotypes under stress and high to moderate performance genotypes under NDS conditions, classified as Group B according to Fernandez [42] had high values of SSI, TOL, and SSPI indices. Thus, they were susceptible to drought and suitable for non-stress conditions only, while the poorest performance genotypes under both stress and non-stress conditions, pooled in group D following Fernandez [42], had the lowest values for STI, HM, GMP, MP, YSI, and Yp. Hence, these genotypes were identified as highly sensitive genotypes to drought with no stability in performance. The majority of these genotypes were found in cluster 3 of the dendrogram for both VDS and FDS. Identifying of sensitive genotypes by clustering accessions with the help of yield-based drought tolerance indices was also performed in *Phaseolus vulgaris* [35] and *Eragrostis tef* genotypes as reported Ferede et al. [67]

## Conclusion

This study showed a decreased in grain yield of 70.50 and 83.60% respectively when cowpea plants are exposed to drought stress at the vegetative and the flowering stages. This severe drought allowed the discrimination of cowpea genotypes in both VDS and FDS. Drought intensity index, Stress susceptibility percentage index, Percentage of yield reduction were significantly lower for VDS compared to FDS and Ys, STI, GMP, MP and DTI were significantly higher for VDS compared to FDS, pointing the drought at the flowering stage being significantly more sensitive and impacting negatively seed yield compared to drought that can take place at the vegetative stage. Based on principal component analysis, the indices STI, REI, GMP, MP, YI and HM exhibited strong correlation with yield under stress and non-stress conditions and are suitable for discriminating high-yielding and drought tolerant genotypes under both stress and non-stress conditions. Based on ranking, PCA and cluster analysis of the studied cowpea genotypes, the top five tolerant to drought stress under VDS were CP-016, CP-021, MTA-22, CP-056 and CP-060 and were CP-016, CP-056, CP-021, CP-028 and MTA-22 under FDS.

## Author Contributions

**Conceptualization:** Eric Bertrand Kouam.

**Data curation:** Toscani Ngompe Deffo, Eric Bertrand Kouam, Marie Solange Mandou, Raba Allah-To Bara.

**Formal analysis:** Toscani Ngompe Deffo, Eric Bertrand Kouam.

**Investigation:** Toscani Ngompe Deffo, Raba Allah-To Bara.

**Methodology:** Toscani Ngompe Deffo, Raba Allah-To Bara.

**Resources:** Christopher Mubeteneh Tankou.

**Software:** Eric Bertrand Kouam.

**Supervision:** Eric Bertrand Kouam, Marie Solange Mandou.

**Validation:** Eric Bertrand Kouam, Marie Solange Mandou, Asafor Henry Chotangui, Adamou Souleymanou, Honore Beyegue Djonko, Christopher Mubeteneh Tankou.

**Writing – original draft:** Toscani Ngompe Deffo.

**Writing – review & editing:** Eric Bertrand Kouam, Marie Solange Mandou, Asafor Henry Chotangui, Adamou Souleymanou, Honore Beyegue Djonko.

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
