## [Decision Letter · Decision Letter 0]

1 Feb 2024

PONE-D-23-38326Yield-based drought tolerance indices in cowpea exposed to stress at vegetative and flowering setting stages identifies sensitive growth phase and drought tolerant genotypes in prospect to crop improvementPLOS ONE

Dear Dr. Kouam,

Thank you for submitting your manuscript to PLOS ONE. After careful consideration, we feel that it has merit but does not fully meet PLOS ONE’s publication criteria as it currently stands. Therefore, we invite you to submit a revised version of the manuscript that addresses the points raised during the review process.

We look forward to receiving your revised manuscript.

Kind regards,

Sumit Jangra, Ph.D.

Academic Editor

PLOS ONE

2. PLOS requires an ORCID iD for the corresponding author in Editorial Manager on papers submitted after December 6th, 2016. Please ensure that you have an ORCID iD and that it is validated in Editorial Manager. To do this, go to ‘Update my Information’ (in the upper left-hand corner of the main menu), and click on the Fetch/Validate link next to the ORCID field. This will take you to the ORCID site and allow you to create a new iD or authenticate a pre-existing iD in Editorial Manager. Please see the following video for instructions on linking an ORCID iD to your Editorial Manager account: " ext-link-type="uri" xlink:type="simple">https://www.youtube.com/watch?v=_xcclfuvtxQ".

3. In the online submission form, you indicated that [Insert text from online submission form here]. 

Reviewers' comments:

Reviewer's Responses to Questions

**Comments to the Author**

1. Is the manuscript technically sound, and do the data support the conclusions?

Reviewer #1: Yes

Reviewer #2: Yes

Reviewer #3: Partly

2. Has the statistical analysis been performed appropriately and rigorously? 

Reviewer #1: Yes

Reviewer #2: No

Reviewer #3: I Don't Know

3. Have the authors made all data underlying the findings in their manuscript fully available?

Reviewer #1: Yes

Reviewer #2: No

Reviewer #3: Yes

4. Is the manuscript presented in an intelligible fashion and written in standard English?

Reviewer #1: Yes

Reviewer #2: No

Reviewer #3: No

5. Review Comments to the Author

Reviewer #1: This study on “Yield-based drought tolerance indices in cowpea exposed to stress at vegetative and flowering setting stages identifies sensitive growth phase and drought tolerant genotypes in prospect to crop improvement” reveals very good information about the 14 drought tolerance indices in response to the drought stress under early vegetative stage and flower setting stage of the cow pea genotypes. This study will help in selection of appropriate drought tolerance index based on the requirement. However, the manuscript has a lot of ambiguous sentences. Sentences and grammar need to be improved. Was there any reason for selection of genotypes from Sahelian and western regions of Cameroon?

Table 3 and 4 are too big to have the view. Authors may find more suitable way of presenting the important information of these two tables, and these tables can be kept as supplementary information.

In the correlation table 5, different drought tolerance indices behave differently with the yield under VDS and FDS conditions. Authors need to make the interpretations of the results keeping in mind the indices and their components of consideration.

Similarly, PCA analyses also gives a very important finding on indices and their association and therefore their appropriate use.

There are some minor improvement needed, and some of which I have mentioned in the attached annotated file.

Reviewer #2: - Please choose the words other than the words present in title and avoid repetition such as word drought.

- Please take care of spacing and grammatical errors throughout MS.

- I suggest adding a map of the study area and providing more details on climatic conditions.

- What is the duration of vegetative and reproductive stage in cowpea? 28 days is not long covering major duration of these stages?

o At least temperature data needs to be added in MS.

o Soil physical properties: particularly texture, field capacity, PAW and PWP must be added.

o Soil moisture data needs to be presented to see soil water changes during stress period.

o How FC was maintained when “Under non-drought stress condition, plants were

regularly watered up to field capacity until physiological maturity.”

o 5-liter pot capacity looks insufficient to support growth for 2 plants till harvest. What are the dimensions of pots? Need to see soil moisture data.

- ANOVA table must be presented before means.

- I suggest replacing Pearson’s table to Correlation plot.

- The following articles might be useful in improving introduction and results and discussion.

o https://doi.org/10.3390/plants10122565

o https://doi.org/10.3390/agronomy13092218

o https://doi.org/10.3390/agronomy12102518

- Please check and correct references

Reviewer #3: The manuscript has lacking in leading the reader towards a specific conclusion. So, the direction of the paper or discussion should be oriented towards the objective of the paper.

-As the paper is completely focused on indices based on yield. So, how the plants responded after imposing the drought treatments is hard to understand only based on yield. Different plants traits during and after treatments, along with their interaction to the grain yield could improve the understanding.

-As the study is based on pot experiment, so how the study results/findings would reflect in the field conditions. Author needs to clarify this.

-drought scenarios to the targeted country or region might be included in the Introduction section.

6. PLOS authors have the option to publish the peer review history of their article (what does this mean?). If published, this will include your full peer review and any attached files.

Reviewer #1: **Yes: **Gayacharan

Reviewer #2: No

Reviewer #3: **Yes: **Md. Ashraful Alam

---

## [Author Response · Author response to Decision Letter 0]

26 Mar 2024

RESPONSES TO REVIEWERS

5. Review Comments to the Author

Reviewer #1: This study on “Yield-based drought tolerance indices in cowpea exposed to stress at vegetative and flowering setting stages identifies sensitive growth phase and drought tolerant genotypes in prospect to crop improvement” reveals very good information about the 14 drought tolerance indices in response to the drought stress under early vegetative stage and flower setting stage of the cow pea genotypes. This study will help in selection of appropriate drought tolerance index based on the requirement. However, the manuscript has a lot of ambiguous sentences. Sentences and grammar need to be improved. 

Sentences and grammar have been improved in the revised manuscript

Was there any reason for selection of genotypes from Sahelian and western regions of Cameroon?

We found judicial to include in our study genotypes from these two regions since cowpea are mainly and significantly produced in these both regions

Table 3 and 4 are too big to have the view. Authors may find more suitable way of presenting the important information of these two tables, and these tables can be kept as supplementary information.

These Tables 3 and 4 present main results of the manuscript. Please I would like these tables to be present as it is in the manuscript

In the correlation table 5, different drought tolerance indices behave differently with the yield under VDS and FDS conditions. Authors need to make the interpretations of the results keeping in mind the indices and their components of consideration. Similarly, PCA analyses also gives a very important finding on indices and their association and therefore their appropriate use.

Thanks for the observations. They were considered in the revised manuscript

There are some minor improvements needed, and some of which I have mentioned in the attached annotated file.

All the recommendations have been considered in the revised manuscript

Reviewer #2: - Please choose the words other than the words present in title and avoid repetition such as word drought.

Amendment was made on the title in the revised manuscript

- Please take care of spacing and grammatical errors throughout MS.

These was done throughout the manuscript

- I suggest adding a map of the study area and providing more details on climatic conditions.

The study site was described with geographical coordinates (longitude and latitude), elevation (altitude), average air temperature, relative humidity and soil characteristics in the revised manuscript

- What is the duration of vegetative and reproductive stage in cowpea? 28 days is not long covering major duration of these stages?

Life cycle in cowpea is in average 60 days for cowpea from the western region and 40 days for the Sahelian accessions. 28 days (4 weeks) is a quite significant within cowpea life cycle

o At least temperature data needs to be added in MS.

Temperature data of the study site is included in the revised manuscript

o Soil physical properties: particularly texture, field capacity, PAW and PWP must be added. o Soil moisture data needs to be presented to see soil water changes during stress period.

 Details of the soil characteristics have been provided in the revised manuscript

o How FC was maintained when “Under non-drought stress condition, plants were

regularly watered up to field capacity until physiological maturity.” o 5-liter pot capacity looks insufficient to support growth for 2 plants till harvest. What are the dimensions of pots? Need to see soil moisture data.

Pots used in the experiment contains holes at the bottom, those pots were place on a plastic plate that received the exceed of water after watering. Then pot was watered until the plastic plate started receiving the exceed of water. This is how the FC was regularly maintained. 

Pot dimensions were added in the revised manuscript. They were 25X30 cm and were sufficient to contain 2 cowpea plants

-. ANOVA table must be presented before means.

ANOVA test was provided in the revised manuscript

- I suggest replacing Pearson’s table to Correlation plot.

Correlation table was replaced by correlation plot in the revised manuscript

- The following articles might be useful in improving introduction and results and discussion.

o https://doi.org/10.3390/plants10122565

o https://doi.org/10.3390/agronomy13092218

o https://doi.org/10.3390/agronomy12102518

These important articles were considered in the revised manuscript

- Please check and correct references

References were checked again and corrections were made in the revised manuscript

Reviewer #3: The manuscript has lacking in leading the reader towards a specific conclusion. So, the direction of the paper or discussion should be oriented towards the objective of the paper.

As stated in the introduction in the revised manuscript, the paper has three clear distinct objectives and conclusions was made concerning those objectives in the revised manuscript

-As the paper is completely focused on indices based on yield. So, how the plants responded after imposing the drought treatments is hard to understand only based on yield. Different plants traits during and after treatments, along with their interaction to the grain yield could improve the understanding. 

Thanks so much for the remark. I appreciate it. We will consider it during the next study using these same genotypes. The current research was only focused on yield unde non-stress and stress conditions and drought tolerance indices

-As the study is based on pot experiment, so how the study results/findings would reflect in the field conditions. Author needs to clarify this.

This study used a huge quantity of plant material (88 accessions) and needed to be carried out in a controlled environment (Greenhouse) to appreciate the variation in the different irrigation regimes and classify genotypes according to their tolerance to drought that are to be reflected in normal field conditions

-drought scenarios to the targeted country or region might be included in the Introduction section.

This have been done in the revised manuscript

---

## [Decision Letter · Decision Letter 1]

10 May 2024

PONE-D-23-38326R1Identifying critical growth stage and resilient genotypes in cowpea under drought stress contributes to enhancing crop tolerance for improvement and adaptation in CameroonPLOS ONE

Dear Dr. Kouam,

Thank you for submitting your manuscript to PLOS ONE. After careful consideration, we feel that it has merit but does not fully meet PLOS ONE’s publication criteria as it currently stands. Therefore, we invite you to submit a revised version of the manuscript that addresses the points raised during the review process.

**Accepted with minor revision**

We look forward to receiving your revised manuscript.

Kind regards,

Sumit Jangra, Ph.D.

Academic Editor

PLOS ONE

Journal Requirements:

Reviewers' comments:

Reviewer's Responses to Questions

**Comments to the Author**

1. If the authors have adequately addressed your comments raised in a previous round of review and you feel that this manuscript is now acceptable for publication, you may indicate that here to bypass the “Comments to the Author” section, enter your conflict of interest statement in the “Confidential to Editor” section, and submit your "Accept" recommendation.

Reviewer #2: All comments have been addressed

Reviewer #4: All comments have been addressed

2. Is the manuscript technically sound, and do the data support the conclusions?

Reviewer #2: Yes

Reviewer #4: Yes

3. Has the statistical analysis been performed appropriately and rigorously? 

Reviewer #2: Yes

Reviewer #4: Yes

4. Have the authors made all data underlying the findings in their manuscript fully available?

Reviewer #2: Yes

Reviewer #4: Yes

5. Is the manuscript presented in an intelligible fashion and written in standard English?

Reviewer #2: Yes

Reviewer #4: Yes

6. Review Comments to the Author

Reviewer #2: Thank you for revising the manuscript.

Just some quick notes to consider,

Please adjust size of tables 3,4 as the last column is cropped.

I suggest to adjust the axis of PCA to lower range: in this way parameters will be more clear.

Reviewer #4: Please add PC3 and PC4 also because author calculated only two like PC1 and PC2 so he also calculated latest like PC3 and PC4 with clealy defined

7. PLOS authors have the option to publish the peer review history of their article (what does this mean?). If published, this will include your full peer review and any attached files.

Reviewer #2: No

Reviewer #4: **Yes: **Dr. Wajid Ali Jatoi

---

## [Author Response · Author response to Decision Letter 1]

14 May 2024

RESPONSES TO REVIEWERS

6. Review Comments to the Author

Reviewer #2: Thank you for revising the manuscript.

Just some quick notes to consider,

Please adjust size of tables 3,4 as the last column is cropped.

Tables 3 and 4 have been adjusted in the revised manuscript. All the columns are now well seen as those tables are now well fit on the pages

I suggest to adjust the axis of PCA to lower range: in this way parameters will be more clear.

The range of the PCA axis was set by the XLSTAT program when performing the analysis

Reviewer #4: Please add PC3 and PC4 also because author calculated only two like PC1 and PC2 so he also calculated latest like PC3 and PC4 with clealy defined

We focussed the principal component analysis on PC1 and PC2 because both explained the essential of the variation that was observed. For VDS PC1 and PC2 cumulated 94.57% of the total variation and the same trend was observed with FDS, PC1 and PC1 cumulating 92.42% of the total variation

---

## [Editor Report · Decision Letter 2]

16 May 2024

Identifying critical growth stage and resilient genotypes in cowpea under drought stress contributes to enhancing crop tolerance for improvement and adaptation in Cameroon

PONE-D-23-38326R2

Dear Dr. Kouam,

We’re pleased to inform you that your manuscript has been judged scientifically suitable for publication and will be formally accepted for publication once it meets all outstanding technical requirements.

Kind regards,

Sumit Jangra, Ph.D.

Academic Editor

PLOS ONE
---

## [Editor Report · Acceptance letter]

19 Jun 2024

PONE-D-23-38326R2 

PLOS ONE

Dear Dr. Kouam, 

I'm pleased to inform you that your manuscript has been deemed suitable for publication in PLOS ONE. Congratulations! Your manuscript is now being handed over to our production team.

Kind regards, 

on behalf of

Dr. Sumit Jangra 

Academic Editor

PLOS ONE